# Integrating Expert ODEs into Neural ODEs: Pharmacology and Disease Progression

**Zhaozhi Qian**
University of Cambridge
zq224@cam.ac.uk

**William R. Zame**
UCLA
zame@econ.ucla.edu

**Lucas M. Fleuren**
Amsterdam UMC
l.fleuren@amsterdamumc.nl

**Paul Elbers**
Amsterdam UMC
p.elbers@amsterdamumc.nl

**Mihaela van der Schaar**
University of Cambridge
UCLA
The Alan Turing Institute
mv472@cam.ac.uk

## Abstract

Modeling a system's temporal behaviour in reaction to external stimuli is a fundamental problem in many areas. Pure Machine Learning (ML) approaches often fail in the small sample regime and cannot provide actionable insights beyond predictions. A promising modification has been to incorporate expert domain knowledge into ML models. The application we consider is predicting the patient health status and disease progression over time, where a wealth of domain knowledge is available from pharmacology. Pharmacological models describe the dynamics of carefully-chosen medically meaningful variables in terms of systems of Ordinary Differential Equations (ODEs). However, these models only describe a limited collection of variables, and these variables are often not observable in clinical environments. To close this gap, we propose the latent hybridisation model (LHM) that integrates a system of expert-designed ODEs with machine-learned Neural ODEs to fully describe the dynamics of the system and to link the expert and latent variables to observable quantities. We evaluated LHM on synthetic data as well as real-world intensive care data of COVID-19 patients. LHM consistently outperforms previous works, especially when few training samples are available such as at the beginning of the pandemic.

## 1  Introduction

Understanding the temporal evolution of a dynamical system is *the* central problem in many areas. The Machine Learning (ML) approach to this problem has been to learn a collection of latent variables and construct a dynamical model of the system directly from observational data. While ML has achieved strong predictive performance in some applications, it has two central weaknesses. The first is that it requires large datasets. The second is that the latent variables that the ML approach identifies often have no physical interpretation and do not correspond to any previously-identified quantities.

One approach to dealing with these weaknesses has been to incorporate expert domain knowledge into ML models. Most of the work using this approach has focused on incorporating *high-level* knowledge about the underlying physical system, such as conservation of energy [9, 32, 99], independence of mechanism [63], monotonicity [59], or linearity [34]. In addition, there have been attempts to integrate domain-specific "expert models" into ML models to create "hybrid" models. Most of this work has employed expert models that directly issue predictions [54, 86, 91, 93] or extract useful features from the raw measurements [41].

35th Conference on Neural Information Processing Systems (NeurIPS 2021).

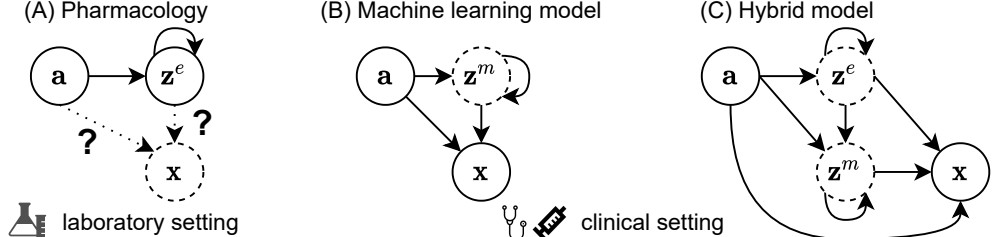

Figure 1: Dependency structure of the three models designed for the laboratory or clinical settings. Dashed nodes represent unobservable variables. The expert variables $\mathbf{z}^e$ are observable in the laboratory setting but not in the clinical setting. The pharmacological model does not contain the links to the clinical variables $\mathbf{x}$.

The approach taken in this paper begins with an expert model in the form of a system of Ordinary Differential Equations (ODEs) and integrates that expert model into a system of Neural ODEs [14]. The specific problem we address is that of predicting disease progression and health status over time; the specific expert model(s) come from Pharmacology [42] – but we believe our approach may be much more widely applicable. For a number of diseases, available pharmacological models, built on the basis of specialized knowledge and laboratory experiments, provide a description of the dynamics of carefully-chosen medically meaningful variables in terms of ODEs that govern the evolution of these states [20, 2, 30]. However, these models are typically not directly applicable in clinical environments, because they involve too few variables to fully describe a patient's health state [80, 38], because the expert variables which the models employ may be observable in the laboratory setting but not in clinical environments [29, 6], and because the relationships between the expert variables and clinically observable quantities is not known [25]. We will give a example later in Section 3.5.

This paper proposes a novel hybrid modeling framework, the Latent Hybridisation Model (LHM), that imbeds a given pharmacological model (a collection of expert variables and the ODEs that describe the evolution of these variables ) into a larger latent variable ML model (a system of Neural ODEs). In the larger model, we use observational data to learn *both* the evolution of the unobservable latent variables and the relationship between measurements and *all* the latent variables – the expert variables from the pharmacological model *and* the latent variables in the larger model. The machine learning component provides links between the expert variables and the clinical measurements, the underlying pharmacological model improves sample efficiency, and the expert variables provide additional insights to the clinicians. A variety of experiments (using synthetic and real data) demonstrate the effectiveness of our hybrid approach.

## 2   Problem setting

We consider a set of hospitalized patients $[N] = \{1, \ldots, N\}$ over a time horizon $[0, T]$; $t = 0$ represents the time of admission and $t = T$ represents the maximal length of stay. The health status of each patient $i$ is characterized by a collection of *observable* physiological variables $\mathbf{x}_i(t) \in \mathbb{R}^D$, $D \in \mathbb{N}^+$; because the physiological variables may include vital signs, bloodwork values, biomarkers, etc., $\mathbf{x}_i(t)$ is typically a high-dimensional vector. Although the physiological variables are observable, they are typically *measured* only at discrete times, and with error. To avoid confusion, we distinguish the measurements of these variables $\mathbf{y}(t)$ from the true values; i.e.

$$\mathbf{y}_i(t) = \mathbf{x}_i(t) + \epsilon_{it} \tag{1}$$

where the independent noise term $\epsilon_{it}$ accommodates the measurement error (modeling $\epsilon_t$ as an autocorrelated stochastic process is left as a future work). For illustrative purposes, we also assume that $\epsilon_t$ follows a Normal distribution $N(0, \sigma_i^2)$, but any parametric distribution could be easily accommodated. We denote the measurement times for each patient as $\mathcal{T}_i = \{t_{i1}, t_{i2}, \ldots\}$. We write $\mathbf{a}_i(t) \in \mathbb{R}^A$, $A \in \mathbb{N}^+$ for the *treatments* the patient receives. Some treatments (e.g. intravenous medications) are continuous; others (e.g. surgical interventions) are discrete, so some components of $\mathbf{a}_i(t)$ may be continuous functions but others are (discontinuous) step functions.

It is convenient to write $\mathcal{A}_i[t_1 : t_2] = \{\mathbf{a}_i(t) | t_1 \leq t \leq t_2\}$ and $\mathcal{Y}_i[t_1 : t_2] = \{\mathbf{y}_i(t) | t_1 \leq t \leq t_2, t \in \mathcal{T}_i\}$ for the treatments and measurements (respectively) during the the time window $[t_1, t_2]$. Note that

$\mathcal{Y}_i[0:t]$ and $\mathcal{A}_i[0:t]$ represent *histories* at time $t$ while $\mathcal{A}_i[t:T]$ and $\mathcal{Y}_i[t:T]$ represent *treatment plans* and *predictions*, respectively. Our objective is to *predict* the future measurements given the history and a treatment plan $\mathcal{A}_i[t_0:T]$:

$$\mathbb{P}(\mathcal{Y}_i[t_0:T] \mid \underbrace{\mathcal{Y}_i[0:t_0], \mathcal{A}_i[0:t_0]}_{\text{Historical observations}}, \underbrace{\mathcal{A}_i[t_0:T]}_{\text{Treatment plan}}). \tag{2}$$

Understanding this distribution will allow us to compute both point estimates and credible intervals (reflecting uncertainty). Note that it is important for the clinician to understand uncertainty in order to balance risk and reward. When the context is clear, we will omit the subscript $i$ and the time index $t$.

## 3 Method

### 3.1 The pharmacological model

We begin with a pharmacological model which describes the dynamics of a collection of "expert" variables $\mathbf{z}^e(t) \in \mathbb{R}^E$. Each expert variable captures a distinct and medically-meaningful aspect of the human body, e.g. the activation of immune system. The pharmacological model describes the dynamics as a system of Ordinary Differential Equations (ODEs):

$$\dot{\mathbf{z}}^e(t) = f^e(\mathbf{z}^e(t), \mathbf{a}(t); \theta^e), \tag{3}$$

where we have written $\dot{\mathbf{z}}^e(t)$ for the time derivative of $\mathbf{z}^e$. The functional form of $f^e : \mathbb{R}^E \times \mathbb{R}^A \to \mathbb{R}^E$ is specified but the unknown parameters $\theta^e$ (e.g., coefficients) need to be estimated from data.[1,2]

It is important to note that the system of ODEs (3) describes dynamics that are *self-contained*, in the sense that the time derivatives $\dot{\mathbf{z}}^e(t)$ depend only on the *current* values of the expert variables $\mathbf{z}^e(t)$ and the *current* treatments $\mathbf{a}(t)$, and not on histories or on other variables. To ensure that this obtains, it may be necessary to limit the scope of the model and limit attention to a single system of the body (or perhaps to several closely related systems) [19]. As a consequence of these limitations, the expert variables will usually not give a full picture of the health status of the patient and will usually not account for the full array of *observable* physiological variables $\mathbf{x}(t)$ [29].

### 3.2 The latent hybridisation model: linking expert variables with measurements

As we have already noted, pharmacological models are typically developed and calibrated in the laboratory, where the expert variables can be directly measured – in patients, in laboratory animals, or even in vitro (Figure 1 A). In clinical environments, the expert variables are frequently not observed (Figure 1 B and C). To use the pharmacological models in clinical environments, we must establish links between the expert variables $\mathbf{z}^e(t)$ and the clinical measurements $\mathbf{y}(t)$. To do this we introduce additional latent variables $\mathbf{z}^m(t) \in \mathbb{R}^M$ and posit the following relationship between the latent variables $\mathbf{z}^e, \mathbf{z}^m$ and the observable physiological variables $\mathbf{x}(t)$:

$$\mathbf{x}(t) = g(\mathbf{z}^e(t), \mathbf{z}^m(t), \mathbf{a}(t); \gamma) \tag{4}$$

The function $g : \mathbb{R}^{E \times M \times A} \to \mathbb{R}^D$ is a neural network with (unknown) weights $\gamma$, and maps the latent space to the "physiological space". Here we also allow the treatment $\mathbf{a}(t)$ to affect the mapping between the latent $\mathbf{z}^e(t), \mathbf{z}^m(t)$ and the observable $\mathbf{x}(t)$. This is a standard design in the state space modeling literature, which we conform to (e.g. Equation 1 in [8] and Equation 1 and 2 in [85]). We posit that the dynamics of the latent variables $\mathbf{z}^m(t)$ follow a system of ODEs governed by its current values $\mathbf{z}^m(t)$, the treatments $\mathbf{a}(t)$ *and* the current values of the expert variables $\mathbf{z}^e(t)$.

$$\dot{\mathbf{z}}^m(t) = f^m(\mathbf{z}^m(t), \mathbf{z}^e(t), \mathbf{a}(t); \theta^m), \tag{5}$$

The function $f^m : \mathbb{R}^{M \times E \times A} \to \mathbb{R}^M$ is a neural network with (unknown) weights $\theta^m$. Equations (3)-(5) specify the dynamics of LHM. It is convenient to write $\mathbf{z}(t) = [\mathbf{z}^e(t) \ \mathbf{z}^m(t)]$ for the vector of all latent variables and $\Theta = (\theta^e, \theta^m, \gamma, \sigma)$ for the set of all (unknown) coefficients.

---

[1] The system in Equation (3) is quite general; appropriate choices of expert variables allow it to capture both high-order ODEs and time-dependent ODEs [67].

[2] Some care must be taken because systems such as (3) do not always admit unique global solutions. In practice, the pharmacological models are sufficiently well-behaved that global solutions exist and are unique. Although closed-form solutions may not be available, there are various efficient numerical methods for solution.

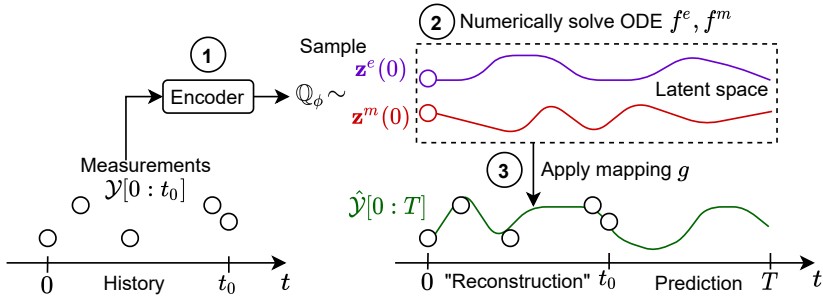

Figure 2: Illustration of the training and prediction procedure.

The coefficients $\Theta$ will be learned from data. However, even after these coefficients are learned, the initial state of the patient $\mathbf{z}_i(0)$ is still *unknown*. In fact, the variation in initial states reflects the heterogeneity of the patient population. If the coefficients and the initial state were known, the entire trajectory of $\mathbf{z}_i$ given the treatments could be computed (numerically). Because we have assumed that the noise/errors $\epsilon_t$ are independent, we have

$$\mathcal{Y}_i[t_0 : T] \perp\!\!\!\perp \mathcal{Y}_i[0 : t_0] \mid \mathbf{z}_i(0), \mathcal{A}_i[0 : T], \Theta, \quad \forall t_0 < T \tag{6}$$

However, because the initial state is unknown, it must be *learned* from the measurements $\mathbf{y}_i(t)$.

LHM would reduce to a pure latent neural ODE model [14, 72] if we omitted the expert variables (Figure 1 B). However, that would amount to *discarding prior (expert) information* and so is evidently undesirable. Indeed, as we have noted in the introduction, our approach is driven by the idea of incorporating this prior (expert) information into our hybrid model.

In the current work, we assume that the pharmacological model in Equation 3 is correct. In practice, the model might be wrong in two ways. The obvious way is that the functional form of $f^e$ might be misspecified (e.g. a linear model might be specified when the truth is actually nonlinear). Many existing techniques can address such misspecification and could be integrated into LHM [35, 64, 98]; see the discussion in Appendix A.7. Alternatively, it might be that the system of expert variables is *not* self-contained, and that their evolution actually depends on additional *latent* variables, we leave this more challenging problem for future work.

Practical extensions to LHM such as including static covariates and modeling informative sampling are discussed in Appendix A.7.

## 3.3 Independent and informative priors

It may be challenging to pinpoint the exact value of the latent variables $\mathbf{z}^e$ based on observations (e.g. due to measurement noise or sampling irregularity). For this reason, we quantify the uncertainty around $\mathbf{z}^e$ using Bayesian inference. In what follows, we assume the initial states $\mathbf{z}_i(0)$ of patients are independently sampled from a prior distribution $\mathbf{z}_i(0) \sim \mathbb{P}_0$. Two points are worth noting.

**Independent Priors.** We use *independent* prior distributions on the expert variables $\mathbf{z}^e$ and the latent variables $\mathbf{z}^m$, i.e. $\mathbb{P}(\mathbf{z}(0)) = \mathbb{P}(\mathbf{z}^e(0)) \times \mathbb{P}(\mathbf{z}^m(0))$. This guarantees that information in $\mathbf{z}^m(0)$ does not duplicate (any of the) information in $\mathbf{z}^e(0)$, which captures our belief that the latent variables are incremental to the expert variables. In addition, independent priors are also commonly used in Bayesian latent variable models such as variational autoencoders (VAEs) [47, 37].

**Informative Priors** The prior distribution on the expert variables $\mathbb{P}(\mathbf{z}^e(0))$ should reflect domain knowledge. Such knowledge is usually available from previous studies in Pharmacology [38]. Using an informative prior tends to improve the estimation of latent variables, especially in small-sample settings [52]. Moreover, the expert variables usually take values in specific ranges (e.g. $[0, 10]$ [42]) and going beyond the valid range may lead to divergence. The informative prior can encode such prior knowledge to stabilize training.

## 3.4 Model training and prediction via amortized variational inference

Given the training dataset $\mathcal{D} = \{(\mathcal{Y}_i[0:T], \mathcal{A}_i[0:T])\}_{i \in [N]}$, we use amortized variational inference (AVI) to estimate the global parameters $\Theta$ and the unknown initial condition $\mathbf{z}_i(0)$ [97]. Figure 2 presents a diagram of the training procedure. We start by learning a *variational distribution* to approximate the posterior $\mathbb{P}(\mathbf{z}_i(0)|\mathcal{Y}_i[0:T], \mathcal{A}_i[0:T])$. As is standard in AVI [47, 97], we use a Normal distribution with diagonal covariance matrix as approximation:

$$\mathbb{Q}(\mathbf{z}_i(0)|\mathcal{Y}_i[0:T], \mathcal{A}_i[0:T]) = N(\mu_i, \Sigma_i); \quad \mu_i, \Sigma_i = e(\mathcal{Y}_i[0:T], \mathcal{A}_i[0:T]; \phi) \tag{7}$$

Here the parameters $\mu_i, \Sigma_i$ are produced by an inference network (also known as an encoder) $e(\cdot)$ with trainable weights $\phi$. When the context is clear, we will denote the variational distribution defined by Equations (7) as $\mathbb{Q}_\phi$. The evidence lower bound (ELBO) for the global parameter $\Theta$ and the inference network parameters $\phi$ is defined as

$$\text{ELBO}(\Theta, \phi) = \mathbb{E}_{\mathbf{z}(0) \sim \mathbb{Q}_\phi}\big[\log\mathbb{P}(\mathcal{Y}_i[0:T]|\mathcal{A}_i[0:T], \mathbf{z}(0), \Theta)\big] - \text{KL}[\mathbb{Q}_\phi|\mathbb{P}_0] \tag{8}$$

To compute the ELBO for a given $\Theta$ and $\phi$, we sample $\mathbf{z}(0) \sim \mathbb{Q}_\phi$ and numerically solve the ODEs to obtain $\mathbf{z}(t), \forall t \in [0, T]$ (Figure 2; Steps 1, 2). Then, we compute the inner log-likelihood function using the mapping $g$ and the noise distribution (Equations (1) and (4)). Finally, we use Monte Carlo sampling to evaluate the KL divergence term: $\mathbb{E}_{\mathbf{z}(0) \sim \mathbb{Q}_\phi}[\log\mathbb{Q}_\phi(\mathbf{z}(0)) - \log\mathbb{P}_0(\mathbf{z}(0))]$. This is because the informative prior $\mathbb{P}_0$ may not have an analytical KL divergence (unlike the standard Normal prior used in previous works [47, 72]). We optimize ELBO by stochastic gradient ascent and update all parameters jointly in an end-to-end manner (detailed in Appendix A.3).

The prediction procedure follows the same steps as illustrated in Figure 2. For a new patient with history $\mathcal{Y}_i[0:t_0], \mathcal{A}_i[0:t_0]$, we first estimate the variational posterior $\mathbb{Q}_\phi$ using the trained encoder. From Equation (6), we can estimate the target distribution in Equation (2) as:

$$\mathbb{E}_{\mathbf{z}(0) \sim \mathbb{Q}_\phi}\big[\mathbb{P}(\mathbf{y}(t)|\mathbf{z}(0), \mathcal{A}_i[0:t_0], \mathcal{A}_i[t_0:T])\big], \ \forall t > t_0. \tag{9}$$

where $\mathcal{A}_i[t_0:T]$ is a future treatment plan. The outer expectation can be approximated by Monte Carlo sampling from $\mathbb{Q}_\phi$ and the inner probability is given by the likelihood function.

**Choice of variational distribution and encoder**. The training procedure above is agnostic to the exact choice of variational distribution and encoder architecture. We choose the Normal distribution to make fair comparisons with the previous works [14, 72] . For the same reason, we use the reversed time-aware LSTM encoder proposed in [14]. In the Appendix A.4, we show additional experiments with more complex variational distributions, i.e. Normalizing Flows [70].

## 3.5 Using LHM to provide clinical decision support

In order for clinicians to optimally treat patients, they need to predict the progression of disease given the treatments. Although machine learning models may demonstrate feature importance [16, 4], they do not uncover the relationships between those features and the underlying pathophysiology. LHM can provide the missing link between clinical observations and disease mechanisms. In combination with clinical reasoning, it can provide treating clinicians with decision support in several complementary ways.

First, LHM can inform the clinicians about the values of the expert variables $\mathbf{z}^e(t)$ that cannot be observed in the clinical environment but are important for prognosis, choice of treatment, and anticipation of complications. For example, understanding and predicting immune response is pivotal when deciding on immunosuppresive therapy in the treatment of COVID-19: an extreme immune response may lead to a potentially fatal cytokine storm [24], but a suppressed immune response may be equally dangerous in case of (secondary) infection [48, 84]. However, because immune response is not directly observable in the clinical environment, clinicians must rely on proxies such as C-reactive protein (CRP) for inflammation [58]; by their very nature, such proxies are noisy and highly imperfect measures of the desired values.

Secondly, LHM can provide the clinician with predictions of the disease progression given the treatments, enabling the clinicians to design the best treatment plan for the patient at hand.

Finally, LHM can bridge the gap between the laboratory and clinical environments, helping to align model output with clinical reasoning, and thus to bring models to the patient bedside and also to foster translational research [78, 28].

# 4 Related works

**Hybrid models**. Hybrid models combine a given expert model with ML [88]. Depending on the type and functionality of the expert model, various approaches have been proposed. *Residual Models* and *Ensembles* use expert models that can issue predictions directly [54, 86, 89, 91, 93]. A Residual Model fits a ML model to the residuals of the expert model while an Ensemble averages the ML and expert predictions. *Feature Extraction* makes use of an expert model that extracts useful features from the measurements [41]; an ML model then uses these features to make predictions. These methods are not suitable for our setting because our expert model is an ODE that governs the latent variables (5); it does not issue predictions of measurements nor does it extract features that the ML model can use. Appendix A.5 Table 2 summarizes these approaches.

ML inspired by physics uses physical laws to guide the design of architectures [77, 96, 82], loss functions [94, 26], and weight initialization [68]. Examples include Hamiltonian neural networks [32, 9, 99], which reflect the conservation of energy. These models utilize general physical laws rather than a specific expert model, and are rather different than the hybrid models discussed above.

**Works involving expert ODEs**. Some recent works also propose to integrate expert ODEs into the learning system. It is worth comparing with their problem settings and proposed methods. [53] assumes that the observations are directly generated by the expert variables $\mathbf{x}(t) = g(\mathbf{z}^e(t))$ (compare with Equation 4 of this work). It does not involve any machine-learned latent variable or neural ODE. Hence, [53] tends to fall short when the expert variables alone are inadequate to predict the observations. The method in [56] differs from LHM in the type of expert ODE used and the way to incorporate machine-learned latent variables. In [56], the expert ODE depends on an *unknown time-varying* parameter $\theta(t)$, i.e. $\dot{\mathbf{z}}^e(t) = f^e(\mathbf{z}^e(t), \theta(t))$ (compare with Equation 3). It treats $\theta(t)$ as a latent variable and uses ML techniques to infer its value. In contrast, LHM assumes that the expert ODE is self-contained, i.e. with no dependency on any unknown time-varying parameters or variables. The method in [51] addresses a different problem setting, where the expert model is mis-specified. It introduces a ML component $f^m$ to *additively* correct for the discrepancy between the expert ODE and the true dynamics $f$, i.e. $f = f^e + f^m$. It thus relates to the residual models discussed above. Finally, the method in [39] deals with discrete time series and incorporates expert *difference equations*. It is nontrivial to generalize the model and the inference algorithm in [39] to the continuous time setting, which this paper focuses on.

**Neural ODEs**. Neural ODEs approximate unknown ODEs by a neural network [14], frequently using standard feed-forward networks. ODE$^2$VAE uses an architecture with identity blocks to approximate second-order ODEs [95] and GRU-ODE uses an architecture inspired by the Gated Recurrent Unit (GRU) [21, 15]. Neural ODEs and extensions have achieved state-of-the-art performance in a variety of problems involving irregularly-sampled time series data [72, 21, 45]. We discuss other approaches to learning unknown ODEs from data in Appendix A.5.

**Mechanistic models**. Mechanistic models are widely applied in sciences such as Physics [81], Epidemiology [92, 31], and Pharmacology [30, 42]. These models use ODEs to describe a system's continuous-time evolution, possibly under external interventions. The dynamics of the system is specified *deterministically* through the governing ODEs; e.g., Equation (3). When the mechanistic models are developed based on experimental data, they may have a causal interpretation [76] (Appendix A.5). LHM focuses on the prediction problem and it leverages the scientific or causal knowledge encoded in the expert ODE $f^e$ to issue principled predictions.

**Latent variable models**. Latent variable models are widely used in disease progression modeling [87, 4]. These models attempt to infer a set of latent variables to predict complex high-dimensional measurements. The latent variables sometimes have high-level interpretations (e.g. cluster membership), but do not usually correspond to any well-defined and clinically meaningful physiological variable. Moreover, without informative priors, the latent variables can usually be identified only up to certain transformations (e.g. permutation of cluster labels [62]). By contrast, LHM involves medically meaningful expert variables driven by known governing equations and following informative priors.

# 5 Experiment and evaluation

Here we present the results of two experimental studies, one with simulated data and one with real data. In both experiments, we study the effect of dexamethasone treatment for

COVID-19 patients. Both studies are modeled on the real-life treatment of COVID-19 patients in the ICU. The implementation of LHM and the experiment code are available at https://github.com/ZhaozhiQIAN/Hybrid-ODE-NeurIPS-2021 or https://github.com/orgs/vanderschaarlab/repositories

## 5.1 Simulation study

In this simulation, we use LHM to predict the results of a single dexamethasone treatment. Each patient $i$ will receive a one-time treatment; with dosage $d_i \sim$ uniform$[0, 10]$ mg and time $s_i \sim$ uniform$[0, 14]$. Our objective is to predict future measurements.

**Datasets**. We generated a variety of datasets to evaluate the model performance under different scenarios. To evaluate how the number of clinical measurements affects performance, we generated datasets with $D = 20, 40$ or $80$ observable physiological variables $\mathbf{x}$. For each dataset, we set the number of un-modeled states $\mathbf{z}^m$ according to the number of variables in $\mathbf{x}$ to be $M = D/10 = 2, 4$ or $8$ (respectively). (We made this choice to reflect the fact that a larger number of physiological variables often necessitates a larger number of un-modeled states.) We consider a time horizon of $T = 14$ days; this is the median length of stay in hospital for Covid-19 patients [69]. After setting $M$ and $D$, we generate the data points within a dataset independently.

We use a pharmacological model adapted from [18] that describes five expert variables ($E = 5$) under dexamethasone treatment for COVID-19 patients. We use the same model in the real-data experiment. We specify the model and the expert variables in Appendix A.1. The un-modeled states $\mathbf{z}^m$ are governed by the nonlinear ODE $f^m$ shown in equation (5); the true physiological variables $\mathbf{x}$ are generated by the function $g$ in Equation (4). The specifications of $f^m$ and $g$ are provided in Appendix A.4. For each patient $i$, each of the components of its initial condition $\mathbf{z}_i(0)$ were independently drawn from an exponential distribution with rate $\lambda = 100$. Measurement noises are drawn independently from $\epsilon_{it} \sim N(0, \sigma^2)$, for $\sigma = 0.2, 0.4$ or $0.8$; Equation (1). We first simulate all daily measurements at $t = 1, 2, \ldots, T$, and then randomly remove measurements with probability $0.5$ to proxy the fact that measurements are made irregularly.

**Prediction task** For a given patient $i$, we use the measurements $\mathcal{Y}[0 : t_0]$ up to some time $t_0$ and the treatment plan for that particular patient to *predict* the future measurements $\mathcal{Y}[t_0 : T]$. (Note that treatment may have occurred prior to time $t_0$ or may be planned following time $t_0$.) To evaluate the performance under different lengths of observed history, we use $t_0 = 5, 10$ or $12$ days. We evaluate the overall prediction accuracy over $t \in [t_0 : T]$ with Root Mean Squared Error (RMSE). We evaluate the uncertainty calibration in the same time window using the Continuous Ranked Probability Score (CRPS). The evaluation metrics for each future time step is shown in Appendix A.4.

**Training and Evaluation.** We partition each dataset into a training set, a validation set, and a testing set. We consider training sets consisting of $N_0 = 10, 100$, or $500$ data points; each validation set has 100 data points and each testing set has 1000 data points.

**Benchmarks**. We compare the performance of our method (LHM) with the performance of four other methods: latent Neural ODE (NODE), the original Pharmacology model (Expert), the residual model (Residual), and the ensemble model (Ensemble) of Expert and NODE, described below. The details of the optimization and hyper-parameter settings are reported in Appendix A.4.

**NODE** involves $Z$ latent variables $\mathbf{z}(t) \in \mathbb{R}^Z$ whose evolutions are described by $\dot{\mathbf{z}}(t) = f^m(\mathbf{z}(t), \mathbf{a}(t); \theta^m)$, where $f^m$ is a neural network with trainable weights $\theta^m$. The number of latent variables $Z$ is a *hyper-parameter* that is set to be greater than $M + E$, which is the number of *true* latent variables.[3] NODE predict the physiological variables as $\hat{\mathbf{y}}_N(t) = g_N(\mathbf{z}(t), \mathbf{a}(t); \gamma_N)$, where $g_N$ is a a neural network with trainable weights $\gamma_N$. **Expert** is given the true governing equation (3), which describes the expert variables $\mathbf{z}^e(t)$, but no un-modeled latent variables. We use a neural network $g_E$ with trainable weights $\gamma_E$ to predict the physiological variables: $\hat{\mathbf{y}}_E(t) = g_E(\mathbf{z}^e(t), \mathbf{a}(t); \gamma_E)$. **Residual** Given a trained Expert model, we calculate its residuals $\mathbf{r}(t) = \mathbf{y}(t) - \hat{\mathbf{y}}_E(t)$. Then a NODE is trained to predict the residuals. The final prediction is $\hat{\mathbf{y}}_E(t) + \hat{\mathbf{r}}(t)$. **Ensemble** makes prediction as $w_{1t}\hat{\mathbf{y}}_N(t) + w_{2t}\hat{\mathbf{y}}_E(t)$, where $\hat{\mathbf{y}}_N(t)$ and $\hat{\mathbf{y}}_E(t)$ are the predictions issued by NODE and Expert respectively. The ensemble weights $w_{1t}$ and $w_{2t}$ are learned on the validation set to minimize the prediction error.

---

[3]We found that the performance of NODE is not sensitive to the exact choice of $Z$ so long as it is sufficiently larger than $M + E$. (This is consistent with findings reported in the literature [22].)

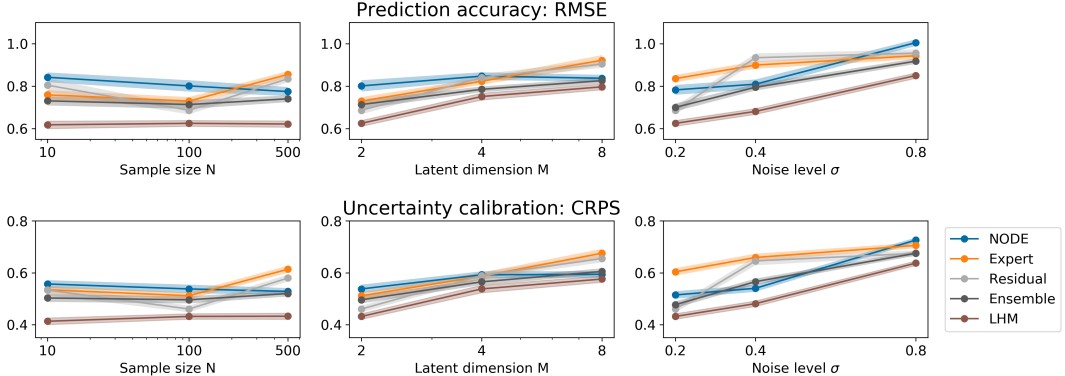

Figure 3: **Simulation Results**. Prediction performance on future measurements $\mathcal{Y}[5:14]$ given the observed history $\mathcal{Y}[0:4]$ as measured by RMSE (first row) and CRPS (second row). The three columns show the results under (1) different training sample sizes $N_0$, (2) different numbers of un-modeled variables $M$, and (3) different noise levels $\sigma$. The shaded areas represent 95% confidence intervals.

**Results.** Figure 3 shows the predictive performance under various samples sizes $N$, number of latent variables $M$, and noise levels $\sigma$ (results for other settings are reported in Appendix A.4). To contextualize the scale of the performance metric, we report that a naive baseline (predicting a constant number of all individuals and all time steps) achieves an RMSE around 1.0. All methods outperform this naive baseline and LHM achieves the best overall performance. Expert does not perform well because it leaves out the latent variables $\mathbf{z}^m(t)$. As a result, its performance does not consistently improve when more training samples become available. In contrast, NODE is flexible and fully data-driven. Although its performance improves with increased sample size, NODE is less sample efficient and it achieves worse performance for all the sample sizes considered. NODE is more robust to the increase in $M$ because it treats the number of latent variables as a hyper-parameter. We observe that the performance gap between NODE and LHM decreases when more latent variables $\mathbf{z}^m(t)$ are added. This is because increasing $M$ reduces the proportion of the variables $E/M$ that can be explained by the expert model. Both Residual and Ensemble achieve performance gains over NODE and Expert alone, but they under-perform LHM because they perform averaging directly in the output space rather than trying to infer $\mathbf{z}^m$ in the latent space.

### 5.2 Real-data experiments

In this experiment, we use real data to evaluate the predictive performance of LHM and to illustrate its utility for decision support in a realistic clinical setting that closely tracks the actual treatment of COVID-19 patients in ICU.

**Dataset**. We used data from the Dutch Data Warehouse (DDW), a multicenter and full-admission anonymized electronic health records database of critically ill COVID-19 patients [27]. Up until March 2021, DDW has collected the health trajectories for 3464 patients admitted to intensive care units (ICU) across the Netherlands. However, even if we use the entire DDW for training, the sample size is still relatively small compared to what is typically used by ML (tens or hundreds of thousands of samples [40]). Furthermore, patients are even scarcer at the early stage of pandemic, arguably when a decision support tool is most needed: only 607 patients were admitted at the first peak (by April 2020). After applying the eligibility criterion detailed in Appendix A.6, we obtained a dataset of **2097** patients whose disease progression is characterized by an irregularly-sampled time series of **27** physiological variables (Appendix A.6). These variables capture the vital signals, respiratory mechanics, and biomarkers that are crucial for clinical decisions. In addition, we also included **11** static variables that are known to affect the progression of COVID-19, e.g. BMI (Appendix A.6).

**Prediction task**. Denote $t_0$ as the time when the patient received the first dose of dexamethasone (we set $t_0 = 24$ for untreated patients). We use the history up to 24 hours before $t_0$, $\mathcal{Y}[t_0 - 24 : t_0]$, to predict the future $\mathcal{Y}[t_0 : t_0 + 24H]$ over a time horizon $H = 1, 3$ or $7$ days. We use $N_0 = 100, 250, 500$ or $1000$ patients for training, 97 for validation, and 1000 for testing. The pharmacological model and the prior distribution of expert variables are detailed in Appendix A.1.

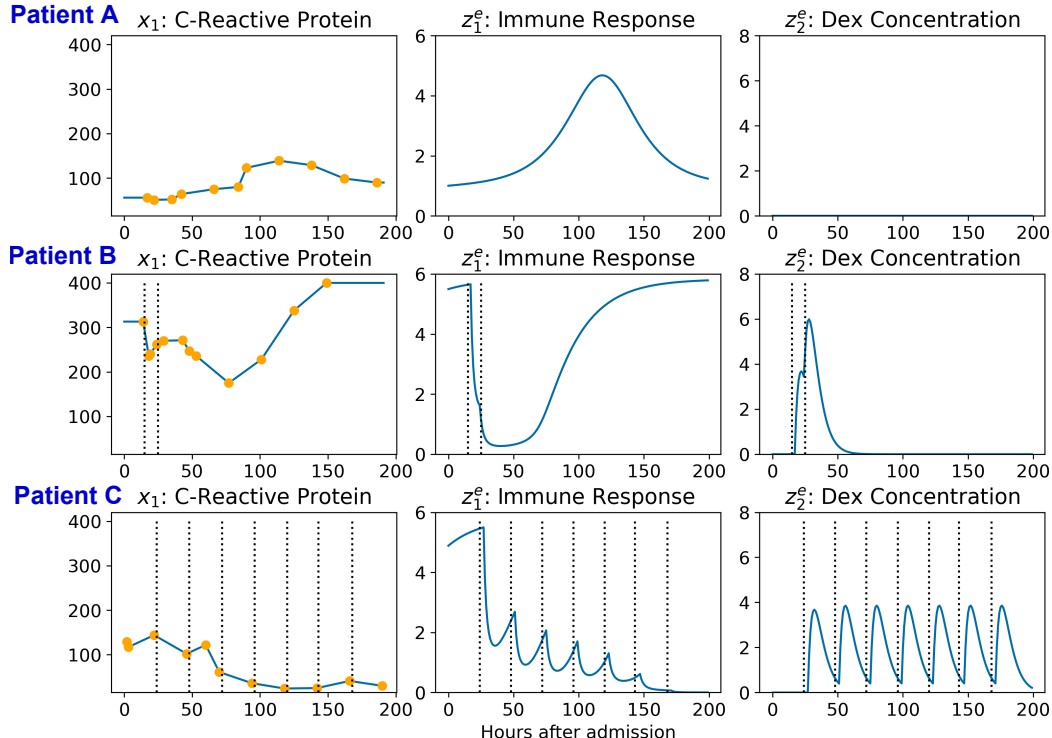

Figure 4: **The observed measurements and the inferred expert variables for three illustrative patients**. Left: The observed physiological variable $x_1(t)$: C-Reactive Protein. Middle: The inferred expert variable $z_1^e(t)$: the immune response to viral infection. Right: The inferred expert variable $z_2^e(t)$: dexamethasone concentration. Vertical dotted lines mark the times of dexamethasone injections.

Table 1: Prediction accuracy (RMSE) on COVID-19 intensive care data under different training sample sizes $N$. Prediction horizon $H = 24$ hours. The standard deviations are shown in the brackets.

| Method $\setminus N_0$ | 100 | 250 | 500 | 1000 |
|---|---|---|---|---|
| Expert | 0.718 (0.71) | 0.704 (0.02) | 0.702 (0.02) | 0.713 (0.01) |
| Residual | 0.958 (0.63) | 1.003 (0.03) | 0.717 (0.05) | 0.635 (0.04) |
| Ensemble | 0.707 (0.60) | 0.657 (0.05) | 0.628 (0.05) | 0.599 (0.05) |
| NODE | 0.662 (0.65) | 0.659 (0.02) | 0.644 (0.05) | 0.650 (0.04) |
| ODE2VAE | 0.674 (0.62) | 0.666 (0.02) | 0.643 (0.02) | 0.619 (0.02) |
| GRU-ODE | 0.722 (0.60) | 0.673 (0.05) | 0.623 (0.05) | 0.601 (0.05) |
| Time LSTM | 0.706 (0.63) | 0.649 (0.03) | 0.600 (0.03) | 0.631 (0.02) |
| LHM | **0.633 (0.51)** | **0.605 (0.02)** | **0.529 (0.02)** | **0.511 (0.02)** |

**Benchmarks**. In addition to all the benchmarks introduced in Section 5.1, we compared the results with two extensions of NODE, GRU-ODE and ODE²VAE, which achieved strong performance in medical time series prediction [95, 21]. We also used the Time LSTM as a strong baseline [7].

**Results**. The main results are shown in Table 1 (additional results are shown in Appendix A.6). LHM consistently outperformed the benchmarks. Its performance with $N_0 = 100$ samples is close to the pure ML approaches' performance with $N_0 = 500$ samples. As the sample size increases from 100 to 1000, the predictive accuracy of LHM improves by 19% while Time LSTM improves by 11% and NODE by less than 5%. A larger improvement rate suggests LHM adapts to the newly available data faster, which is important when the samples are scarce. As expected, the standalone expert model achieved poor performance because it is unable to capture the full array of clinical measurements.

**LHM in action**. Here we show how LHM can support clinical decisions beyond predicting future clinical measurements. Managing the level of immune response is pivotal when deciding on immunosuppresive therapy for COVID-19 patients [24, 84, 48]. This is a challenging task because

the "right" level of immune response varies across patients [43]: for most patients, we would like to reduce the immune response to avoid cytokine storm and consequent organ failure [24], but for patients with other infections (e.g., a secondary bacterial or fungal infection), we would like to keep their immune systems activated [48, 17]. Because immune response is not directly observable in the clinical settings, clinicians resort to unspecific inflammatory markers such as C-Reactive Protein (CRP) [58]. However, better markers of the immune response such as the cytokine Type I IFNs can be measured in the laboratory setting and have been included in the pharmacological model as an expert variable $z_1^e$. Moreover, the immune response is affected by dexamethasone concentrations in the lung tissue (the expert variable $z_2^e$). These concentrations are not easily or routinely measured in a clinical setting, and are therefore not available to clinicians.

Figure 4 shows the *measurements* of CRP $\mathbf{x}_1$, and the *inferred* immune response $z_1^e$ and dexamethasone concentration $z_2^e$. The two expert variables are inferred by a trained LHM using the first five days of observations. We selected three representative patients based on the treatment regimen they received. Patient A is representative of the 59.8% of the population who did not receive dexamethasone; Patient C is representative of the 12.2% who received dexamethasone according to the guidelines [60], and Patient B is representative of the 28.0% of the population who received dexamethasone but whose treatment was not according to the guidelines.

For patient A, the initial level of CRP was moderately high, but then it rose and peaked at about 100 hours after admission. In the absence of contraindications, a clinician might begin dexamethasone treatment at this point, but LHM predicts that immune activity $z_1^e$ would decrease afterwards even without treatment. (The right panel is blank because dexamethasone was never administered.)

Patient B was admitted to the ICU with a very high level of the inflammatory marker CRP. Two doses of dexamethasone were given in rapid succession, preceding a decline in both CRP $x_1$ and immune activity $z_1^e$. However, after the dexamethasone depleted in the patient's body, the expert model predicts that the immune response will pick up again. This is reflected in the re-occurrence of the high CRP level.

Patient C was admitted to the ICU with a moderately high level of CRP. LHM also inferred a high level of immune activity $z_1^e$ at the initial stage. Inflammation was greatly reduced after dexamethasone treatments, which has an immunosuppressive effect; so $x_1$ and $z_1^e$ display the same downward trend. Because dexamethasone concentrations falls rapidly within 24 hours after treatment, it was repeated at 24 hour intervals, as seen in the right panel. This pattern is clinically expected under the treatment regimen.

## 6  Discussion and future work

This paper has focused on a single disease (COVID-19), a single treatment (dexamethasone), and a single expert model. The ultimate goal is to build a model that encompasses a variety of diseases, a variety of treatments and multiple expert models. This is a challenge for future work.

## Acknowledgments and Disclosure of Funding

This work was supported by the US Office of Naval Research (ONR) and the National Science Foundation (NSF, Grant number:1722516).

We would like to thank anonymous reviewers as well as members of the vanderschaar-lab for many insightful comments and suggestions.

We would like to thank the following individuals for providing domain expertise in Pharmacology and designing the expert ODEs.

Bernhard Steiert, PhD, Pharmaceutical Sciences, Roche Pharma Research and Early Development (pRED), Roche Innovation Center Basel, Basel, Switzerland

Richard Peck, MA, MB, BChir, FFPM, FRCP, Pharmaceutical Sciences, Roche Pharma Research and Early Development (pRED), Roche Innovation Center Basel, Basel, Switzerland

We would like to thank the following individuals for curating and sharing the Dutch Data Warehouse (DDW) for critically ill COVID-19 patients.

*From collaborating hospitals having shared data:*

Diederik Gommers, MD, PhD, Department of Intensive Care, Erasmus Medical Center, Rotterdam, The Netherlands,

Olaf L. Cremer, MD, PhD, Intensive Care, UMC Utrecht, Utrecht, The Netherlands,

Rob J. Bosman, MD, ICU, OLVG, Amsterdam, The Netherlands,

Sander Rigter, MD, Department of Anesthesiology and Intensive Care, St. Antonius Hospital, Nieuwegein, The Netherlands,

Evert-Jan Wils, MD, PhD, Department of Intensive Care, Franciscus Gasthuis & Vlietland, Rotterdam, The Netherlands,

Tim Frenzel, MD, PhD, Department of Intensive Care Medicine, Radboud University Medical Center, Nijmegen, The Netherlands,

Dave A. Dongelmans, MD, PhD, Department of Intensive Care Medicine, Amsterdam UMC, Amsterdam, The Netherlands,

Remko de Jong, MD, Intensive Care, Bovenij Ziekenhuis, Amsterdam, The Netherlands,

Marco Peters, MD, Intensive Care, Canisius Wilhelmina Ziekenhuis, Nijmegen, The Netherlands,

Marlijn J.A Kamps, MD, Intensive Care, Catharina Ziekenhuis Eindhoven, Eindhoven, The Netherlands,

Dharmanand Ramnarain, MD, Department of Intensive Care, ETZ Tilburg, Tilburg, The Netherlands,

Ralph Nowitzky, MD, Intensive Care, HagaZiekenhuis, Den Haag, The Netherlands,

Fleur G.C.A. Nooteboom, MD, Intensive Care, Laurentius Ziekenhuis, Roermond, The Netherlands,

Wouter de Ruijter, MD, PhD, Department of Intensive Care Medicine, Northwest Clinics, Alkmaar, The Netherlands,

Louise C. Urlings-Strop, MD, PhD, Intensive Care, Reinier de Graaf Gasthuis, Delft, The Netherlands,

Ellen G.M. Smit, MD, Intensive Care, Spaarne Gasthuis, Haarlem en Hoofddorp, The Netherlands,

D. Jannet Mehagnoul-Schipper, MD, PhD, Intensive Care, VieCuri Medisch Centrum, Venlo, The Netherlands,

Julia Koeter, MD, Intensive Care, Canisius Wilhelmina Ziekenhuis, Nijmegen, The Netherlands,

Tom Dormans, MD, PhD, Intensive care, Zuyderland MC, Heerlen, The Netherlands,

Cornelis P.C. de Jager, MD, PhD, Department of Intensive Care, Jeroen Bosch Ziekenhuis, Den Bosch, The Netherlands,

Stefaan H.A. Hendriks, MD, Intensive Care, Albert Schweitzerziekenhuis, Dordrecht, The Netherlands,

Sefanja Achterberg, MD, PhD, ICU, Haaglanden Medisch Centrum, Den Haag, The Netherlands,

Evelien Oostdijk, MD, PhD, ICU, Maasstad Ziekenhuis Rotterdam, Rotterdam, The Netherlands,

Auke C. Reidinga, MD, ICU, SEH, BWC, Martiniziekenhuis, Groningen, The Netherlands,

Barbara Festen-Spanjer, MD, Intensive Care, Ziekenhuis Gelderse Vallei, Ede, The Netherlands,

Gert B. Brunnekreef, MD, Department of Intensive Care, Ziekenhuisgroep Twente, Almelo, The Netherlands,

Alexander D. Cornet, MD, PhD, FRCP, Department of Intensive Care, Medisch Spectrum Twente, Enschede, The Netherlands,

Walter van den Tempel, MD, Department of Intensive Care, Ikazia Ziekenhuis Rotterdam, Rotterdam, The Netherlands,

Age D. Boelens, MD, Anesthesiology, Antonius Ziekenhuis Sneek, Sneek, The Netherlands,

Peter Koetsier, MD, Intensive Care, Medisch Centrum Leeuwarden, Leeuwarden, The Netherlands,

Judith Lens, MD, ICU, IJsselland Ziekenhuis, Capelle aan den IJssel, The Netherlands,

Roger van Rietschote, Business Intelligence, Haaglanden MC, Den Haag,The Netherlands,

Harald J. Faber, MD, ICU, WZA, Assen, The Netherlands,

A. Karakus, MD, Department of Intensive Care, Diakonessenhuis Hospital, Utrecht, The Netherlands,

Robert Entjes, MD, Department of Intensive Care, Admiraal De Ruyter Ziekenhuis, Goes, The Netherlands,

Paul de Jong, MD, Department of Anesthesia and Intensive Care, Slingeland Ziekenhuis, Doetinchem, The Netherlands,

Thijs C.D. Rettig, MD, PhD, Department of Intensive Care, Amphia Ziekenhuis, Breda, The Netherlands,

M.C. Reuland, MD, Department of Intensive Care Medicine, Amsterdam UMC, Universiteit van Amsterdam, Amsterdam, The Netherlands,

Laura van Manen, MD, Department of Intensive Care, BovenIJ Ziekenhuis, Amsterdam, The Netherlands,

Leon Montenij, MD, PhD, Department of Anesthesiology, Pain Management and Intensive Care, Catharina Ziekenhuis Eindhoven, Eindhoven, The Netherlands,

Jasper van Bommel, MD, PhD, Department of Intensive Care, Erasmus Medical Center, Rotterdam, The Netherlands,

Roy van den Berg, Department of Intensive Care, ETZ Tilburg, Tilburg, The Netherlands,

Ellen van Geest, Department of ICMT, Haga Ziekenhuis, Den Haag, The Netherlands,

Anisa Hana, MD, PhD, Intensive Care, Laurentius Ziekenhuis, Roermond, The Netherlands,

B. van den Bogaard, MD, PhD, ICU, OLVG, Amsterdam, The Netherlands,

Prof. Peter Pickkers, Department of Intensive Care Medicine, Radboud University Medical Centre, Nijmegen, The Netherlands,

Pim van der Heiden, MD, PhD, Intensive Care, Reinier de Graaf Gasthuis, Delft, The Netherlands,

Claudia (C.W.) van Gemeren, MD, Intensive Care, Spaarne Gasthuis, Haarlem en Hoofddorp, The Netherlands,

Arend Jan Meinders, MD, Department of Internal Medicine and Intensive Care, St Antonius Hospital, Nieuwegein, The Netherlands,

Martha de Bruin, MD, Department of Intensive Care, Franciscus Gasthuis & Vlietland, Rotterdam, The Netherlands,

Emma Rademaker, MD, MSc, Department of Intensive Care, UMC Utrecht, Utrecht, The Netherlands,

Frits H.M. van Osch, PhD, Department of Clinical Epidemiology, VieCuri Medisch Centrum, Venlo, The Netherlands,

Martijn de Kruif, MD, PhD, Department of Pulmonology, Zuyderland MC, Heerlen, The Netherlands,

Nicolas Schroten, MD, Intensive Care, Albert Schweitzerziekenhuis, Dordrecht, The Netherlands,

Klaas Sierk Arnold, MD, Anesthesiology, Antonius Ziekenhuis Sneek, Sneek, The Netherlands,

J.W. Fijen, MD, PhD, Department of Intensive Care, Diakonessenhuis Hospital, Utrecht, The Netherland,

Jacomar J.M. van Koesveld, MD, ICU, IJsselland Ziekenhuis, Capelle aan den IJssel, The Netherlands,

Koen S. Simons, MD, PhD, Department of Intensive Care, Jeroen Bosch Ziekenhuis, Den Bosch, The Netherlands,

Joost Labout, MD, PhD, ICU, Maasstad Ziekenhuis Rotterdam, The Netherlands,

Bart van de Gaauw, MD, Martiniziekenhuis, Groningen, The Netherlands,

Michael Kuiper, Intensive Care, Medisch Centrum Leeuwarden, Leeuwarden, The Netherlands,

Albertus Beishuizen, MD, PhD, Department of Intensive Care, Medisch Spectrum Twente, Enschede, The Netherlands,

Dennis Geutjes, Department of Information Technology, Slingeland Ziekenhuis, Doetinchem, The Netherlands,

Johan Lutisan, MD, ICU, WZA, Assen, The Netherlands,

Bart P. Grady, MD, PhD, Department of Intensive Care, Ziekenhuisgroep Twente, Almelo, The Netherlands,

Remko van den Akker, Intensive Care, Adrz, Goes, The Netherlands,

Sesmu Arbous, MD, PhD, Intensivist, LUMC, Leiden, The Netherlands,

Tom A. Rijpstra, MD, Department of Intensive Care, Amphia Ziekenhuis, Breda, The Netherlands,

Roos Renckens, MD, PhD, Department of Internal Medicine, Northwest Clinics, Alkmaar, the Netherlands,

*From collaborating hospitals having signed the data sharing agreement:*

Daniël Pretorius, MD, Department of Intensive Care Medicine, Hospital St Jansdal, Harderwijk, The Netherlands,

Menno Beukema, MD, Department of Intensive Care, Streekziekenhuis Koningin Beatrix, Winterswijk, The Netherlands,

Bram Simons, MD, Intensive Care, Bravis Ziekenhuis, Bergen op Zoom en Roosendaal, The Netherlands,

A.A. Rijkeboer, MD, ICU, Flevoziekenhuis, Almere, The Netherlands,

Marcel Aries, MD, PhD, MUMC+, University Maastricht, Maastricht, The Netherlands,

Niels C. Gritters van den Oever, MD, Intensive Care, Treant Zorggroep, Emmen, The Netherlands,

Martijn van Tellingen, MD, EDIC, Department of Intensive Care Medicine, afdeling Intensive Care, ziekenhuis Tjongerschans, Heerenveen, The Netherlands,

Annemieke Dijkstra, MD, Department of Intensive Care Medicine, Het Van Weel-Bethesda Ziekenhuis, Dirksland, The Netherlands,

Rutger van Raalte, Department of Intensive Care, Tergooi hospital, Hilversum, The Netherlands,

*From the Laboratory for Critical Care Computational Intelligence:*

Tariq A. Dam, MD, Department of Intensive Care Medicine, Laboratory for Critical Care Computational Intelligence, Amsterdam Medical Data Science, Amsterdam UMC, Vrije Universiteit, Amsterdam, The Netherlands,

Martin E. Haan, MD, Department of Intensive Care Medicine, Laboratory for Critical Care Computational Intelligence, Amsterdam Medical Data Science, Amsterdam UMC, Vrije Universiteit, Amsterdam, The Netherlands,

Mark Hoogendoorn, PhD, Quantitative Data Analytics Group, Department of Computer Science, Faculty of Science, VU University, Amsterdam, The Netherlands,

Armand R.J. Girbes, MD, PhD, EDIC, Department of Intensive Care Medicine, Laboratory for Critical Care Computational Intelligence, Amsterdam Medical Data Science, Amsterdam UMC, Vrije Universiteit, Amsterdam, The Netherlands,

Paul W.G. Elbers, MD, PhD, EDIC, Department of Intensive Care Medicine, Laboratory for Critical Care Computational Intelligence, Amsterdam Medical Data Science, Amsterdam UMC, Vrije Universiteit, Amsterdam, The Netherlands,

Patrick J. Thoral, MD, EDIC, Department of Intensive Care Medicine, Laboratory for Critical Care Computational Intelligence, Amsterdam Medical Data Science, Amsterdam UMC, Vrije Universiteit, Amsterdam, The Netherlands,

Dagmar M. Ouweneel, PhD, Department of Intensive Care Medicine, Laboratory for Critical Care Computational Intelligence, Amsterdam Medical Data Science, Amsterdam UMC, Vrije Universiteit, Amsterdam, The Netherlands,

Ronald Driessen, Department of Intensive Care Medicine, Laboratory for Critical Care Computational Intelligence, Amsterdam Medical Data Science, Amsterdam UMC, Vrije Universiteit, Amsterdam, The Netherlands,

Jan Peppink, Department of Intensive Care Medicine, Laboratory for Critical Care Computational Intelligence, Amsterdam Medical Data Science, Amsterdam UMC, Vrije Universiteit, Amsterdam, The Netherlands,

H.J. de Grooth, MD, PhD, Department of Intensive Care Medicine, Laboratory for Critical Care Computational Intelligence, Amsterdam Medical Data Science, Amsterdam UMC, Vrije Universiteit, Amsterdam, The Netherlands,

*From Pacmed:*

Robbert C.A. Lalisang, MD, Pacmed, Amsterdam, The Netherlands,

Michele Tonutti, MRes, Pacmed, Amsterdam, The Netherlands,

Daan P. de Bruin, MSc, Pacmed, Amsterdam, The Netherlands,

Sebastiaan J.J. Vonk, MSc, Pacmed, Amsterdam, The Netherlands,

Mattia Fornasa, PhD, Pacmed, Amsterdam, The Netherlands,

Tomas Machado, Pacmed, Amsterdam, The Netherlands,

Michael de Neree tot Babberich, Pacmed, Amsterdam, The Netherlands,

Olivier Thijssens, MSc, Pacmed, Amsterdam, The Netherlands,

Lot Wagemakers, Pacmed, Amsterdam, The Netherlands,

Hilde G.A. van der Pol, Pacmed, Amsterdam, The Netherlands,

Tom Hendriks, Pacmed, Amsterdam, The Netherlands,

Julie Berend, Pacmed, Amsterdam, The Netherlands,

Virginia Ceni Silva, Pacmed, Amsterdam, The Netherlands,

Robert F.J. Kullberg, MD, Pacmed, Amsterdam, The Netherlands,

Taco Houwert, MSc, Pacmed, Amsterdam, The Netherlands,

Hidde Hovenkamp, MSc, Pacmed, Amsterdam, The Netherlands,

Roberto Noorduijn Londono, MSc, Pacmed, Amsterdam, The Netherlands,

Davide Quintarelli, MSc, Pacmed, Amsterdam, The Netherlands,

Martijn G. Scholtemeijer, MD, Pacmed, Amsterdam, The Netherlands,

Aletta A. de Beer, MSc, Pacmed, Amsterdam, The Netherlands,

Giovanni Cina, PhD, Pacmed, Amsterdam, The Netherlands,

Willem E. Herter, BSc, Pacmed, Amsterdam, The Netherlands,

Adam Izdebski, Pacmed, Amsterdam, The Netherlands,

*From RCCnet:*

Leo Heunks, MD, PhD, Department of Intensive Care Medicine, Amsterdam Medical Data Science, Amsterdam UMC, Vrije Universiteit, Amsterdam, The Netherlands,

Nicole Juffermans, MD, PhD, ICU, OLVG, Amsterdam, The Netherlands,

Arjen J.C. Slooter, MD, PhD, Department of Intensive Care Medicine, UMC Utrecht, Utrecht University, Utrecht, the Netherlands,

*From other collaborating partners:*

Martijn Beudel, MD, PhD, Department of Neurology, Amsterdam UMC, Universiteit van Amsterdam, Amsterdam, The Netherlands,

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
