# A  Appendix

**Summary of Appendices.**

Each section can be read independently.

## A.1  The pharmacological model for dexamethasone

The expert ODE we used is adapted from [18]. As illustrated in Figure 5, it involves five expert variables $z_1$ to $z_5$ (the superscript $e$ is omitted for brevity). The $z_1$ represents the innate immune response to viral infection (measured in the laboratory using Type I IFNs [44, 1]). The $z_2$ and $z_3$ represent the concentration of dexamethasone in lung tissue and plasma respectively. The $z_4$ represents the viral load and $z_5$ represents the adaptive immune response (measured in the laboratory using Cytotoxic T cells [1]).

The expert model that describes these variables are developed based on specialized knowledge and laboratory experiments. Firstly, the immune responses and viral replication are modeled as:

$$\dot{z_1} = k_{IR} \cdot z_4 + k_{PF} \cdot z_4 \cdot z_1 - k_O \cdot z_1 + \frac{E_{max} \cdot z_1^{h_P}}{EC_{50}^{h_P} + z_1^{h_P}} - k_{Dex} \cdot z_1 \cdot z_2 \tag{10}$$

$$\dot{z_4} = k_{DP} \cdot z_4 - k_{IIR} \cdot z_4 \cdot z_1 - k_{DC} \cdot z_4 \cdot z_5^{h_C} \tag{11}$$

$$\dot{z_5} = k_1 \cdot z_1 \tag{12}$$

The five terms in the first Equation for $z_1$ captures the initial immune reaction to the virus, the physiological positive feedback, the immune cell mortality, the pathological positive feedback, and the effect of dexamethasone [18]. The second equation of $z_4$ captures the viral replication, and the effect of innate and adaptive immune systems on the virus. The last equation captures the adaptive immune response triggered by the innate immune response [1]. The unknown coefficients $k_{IR}$, $k_{PF}$, $k_O$, $E_{max}$, $h_P$, $k_{Dex}$, $k_{DP}$, $k_{IIR}$, $k_{DC}$, $h_C$ are positive real numbers.

The concentration of dexamethasone ($z_2$, $z_3$) is described by a standard two-compartmental pharmacokinetics model [55, 49]:

$$\dot{z_2} = -k_2 \cdot z_2 + k_3 \cdot z_3 \tag{13}$$

$$\dot{z_3} = -k_3 \cdot z_3 \tag{14}$$

The coefficients $k_2$, $k_3$ are positive real numbers. In the literature, it is often assumed for simplicity that the treatment is given at time $t = 0$, and the initial condition of the plasma concentration $z_3(0)$ corresponds to the dosage [23]. Since the plasma concentration $z_3$ decays exponentially over time, we can equivalently express it as a sum of exponentials: $z_3(t) = \sum_i d_i \cdot I(t > t_i) \cdot \exp(k_3(t_i - t))$ when dosages $d_i$ are given at time $t_i$, $i \geq 1$. The function $I(\cdot)$ is an indicator function.

**Prior distribution for real-data experiment**. The initial condition $\mathbf{z}(0)$ corresponds to the patient state at the time of ICU admission. Since dexamethasone is generally administered *during* the ICU stay [60], its concentration at admission should be very close to zero. Hence we use an exponential distribution with rate $\lambda = 100$ as the prior of $z_2(0)$ and $z_3(0)$. On the other hand, the immune response and viral load may vary across patients greatly. To allows for more heterogeneity, we use an exponential distribution with rate $\lambda = 0.1$ as the prior of $z_1(0)$, $z_4(0)$, and $z_5(0)$. The exponential distribution also reflects the positivity of the expert variables because it has a positive support.

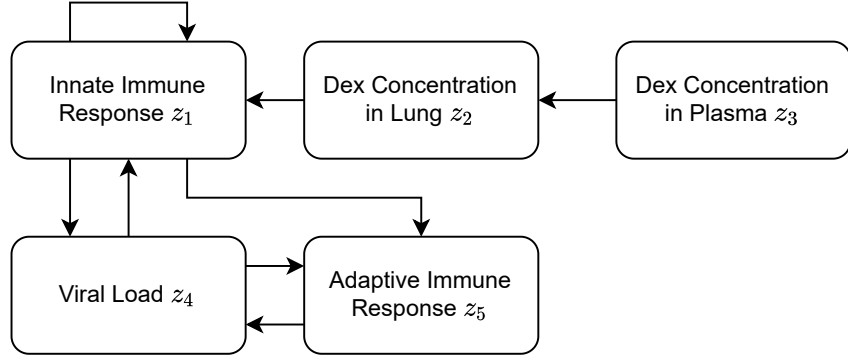

Figure 5: The expert variables and their temporal interactions as described by the expert ODE.

## A.2 Medical importance and impact

### A.2.1 Medical importance of LHM

Integration of machine learning (ML) and pathophysiology is a major challenge in adopting ML models in a clinical setting. While clinicians seek to understand the mechanisms that drive disease progression for prognosis and treatment allocation, machine learning models do not currently provide such disease dynamics. With these dynamics, however, model results would translate to clinically interpretable concepts, would resonate with clinicians, and could then support clinical decision making.

In addition, disease dynamics are indispensable for the clinical interpretation of predictive modeling results. Predictive modeling has taken flight in the medical field, but many models are left stranded because clinicians can solely rely on feature importance and no mechanistic interpretation of the results. Such an interpretation, however, will increase clinicians' trust in these models and expedite their use in clinical practice.

Lastly, the relationship between fundamental and clinical research may yield novel hypotheses and foster subsequent research. There is a gap between benchwork and the bedside. Bridging this gap with ML and interpretable models could reveal novel relationships and could inspire research both ways. Overall, ML with a mechanistic interpretation can provide the next big step in medical data science and can help bring these models to the bedside.

### A.2.2 Dexamethasone in COVID-19

Coronavirus disease 2019 (COVID-19) was an unknown disease to intensive care clinicians worldwide. Both the natural course of the disease as well as optimal treatment were unknown throughout the onset of the pandemic. Since inflammatory organ injury appeared to play an important role in the pathophysiology of COVID-19, glucocorticoids were proposed to mitigate the damaging effects of the immune system [57]. In particular, Dexamethasone treatment has been shown to reduce mortality in patients on invasive mechanical ventilation or oxygen alone in the RECOVERY trial [33]. Moreover, the CoDEX trial demonstrated an increase in the number of ventilator free days with Dexamethasone treatment in moderate to severe COVID-19 acute respiratory distress syndrome (ARDS)[83]. As a result, COVID-19 treatment guidelines recommend Dexamethasone treatment in these settings [60].

Although beneficial effects have been shown of Dexamethasone on a group level, individual response to treatment remains unknown. Knowing this response would help clinicians to anticipate complications, to improve individualized prognosis, and potentially determine beneficial treatments in these patients. Moreover, clinicians could identify patients in which Dexamethasone has a desired effect and in which patient it may not. For example, in the case of coinfection, Dexamethasone may be discontinued in selected patients. Lastly, these models can identify novel mechanistic pathways in COVID-19 patients that can inspire both fundamental and clinical research. Taken together, individualized disease progression in response to Dexamethasone treatment would bring about a large step forward in COVID-19 research.

### A.2.3 Potential negative impact

Any decision support system could be used negatively if the user intentionally chooses to worsen the outcome. This is very unlikely in our case because the intended users of LHM are clinicians.

### A.3 Optimization and gradient calculation

We optimize ELBO by stochastic gradient ascent using the ADAM optimizer [46]. The gradient calculation is enabled by the following two methods.

**Reparameterization**. To evaluate the ELBO, we need to take samples from the variational distribution $\mathbb{Q}_\phi$. Here we use the Gaussian reparameterization in all sampling steps to obtain the gradients for the encoder [47].

**Gradient for ODE**. We use the torchdiffeq library to calculate the gradient with respect to the ODE solutions [14]. A variety of ODE solvers are available in the library, we used the adams solver, which is an adaptive step size solver.

### A.4 Simulation study

#### A.4.1 Data generation details

We generated a variety of datasets to evaluate the model performance under different scenarios. To evaluate how the number of clinical measurements affects performance, we generated datasets with $D = 20, 40$ or $80$ measurable physiological variables $\mathbf{x}$. For the pharmacological model, we used the model provided in Appendix A.1, which involves five inter-related variables ($E = 5$). We set the coefficients $h_P = h_C = 2$ and the rest to be one.

For each dataset, we set the number of un-modeled states $\mathbf{z}^m$ according to the number of observed physiological variables to be $M = D/10 = 2, 4$ or $8$ (respectively). (We made this choice to reflect the fact that a larger number of physiological variables often necessitates a larger number un-modeled states.) The un-modeled states $\mathbf{z}^m$ are governed by a nonlinear ODE
$$\dot{\mathbf{z}}_i^m = \tanh(\mathbf{W}_1 \mathbf{z}_i^m + \mathbf{W}_2 \mathbf{z}_i^e),$$
with the coefficient matrices $\mathbf{W}_1 \in \mathbb{R}^{M \times M}$, $\mathbf{W}_2 \in \mathbb{R}^{M \times E}$. For each dataset, we sampled the entries in these matrices independently from $N(0, 1)$.

For each patient $i$, each of the components of its initial condition $\mathbf{z}_i(0)$ were independently drawn from an exponential distribution with rate $\lambda = 100$ (this distribution is also given to the algorithms as the prior distribution). We consider a time horizon of $T = 14$ days; this is the median length of stay in hospital for Covid-19 patients [69].

Each patient $i$ will receive a one-time dexamethasone treatment with dosage $d_i$ at some time $s_i$, where $d_i \sim \mathrm{uniform}[0, 10]$ mg and $s_i \sim \mathrm{uniform}[0, T]$.

The true physiological variables are generated by
$$\mathbf{x}_i = \mathbf{W}_3 \mathbf{z}_i + \mathbf{W}_4 \mathbf{a}_i,$$
with the coefficient matrices $\mathbf{W}_3 \in \mathbb{R}^{X \times (M+E)}$, $\mathbf{W}_4 \in \mathbb{R}^{X \times 1}$. For each dataset, each element in these matrices was drawn independently from $N(0, 1)$ and then multiplied by a Bernoulli variable with $p = 0.5$, so that approximately half of the elements in each of these matrices $\mathbf{W}_3, \mathbf{W}_4$ were 0. (We did this in order to reflect the idea that each physiological variable is only related to some of the latent variables.) The measurements are generated by
$$\mathbf{y}_i(t) = \mathbf{x}_i(t) + \epsilon_{it}$$
with the measurement noise $\epsilon_{it} \sim N(0, \sigma)$ for $\sigma = 0.2, 0.4$ or $0.8$; Equation (1). We first simulate all the daily measurements at $t = 1, 2, \ldots, T$, and then randomly remove measurements with probability 0.5; this represents the fact that measurements of made irregularly.

#### A.4.2 Hyper-parameter settings

As a reminder, the number of measured clinical variables is $D$, the number of expert variables is $E$, the number of ML latent variables is $M$. The sample size is $N_0$

The following is the hyperparameter setting used in the simulation study:

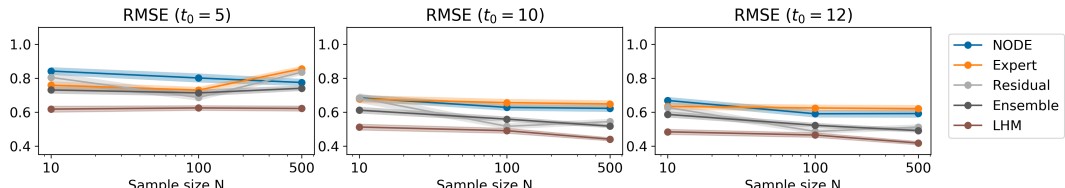

Figure 6: **Simulation results under different lengths of observed history** $t_0$. Prediction accuracy on future measurements $\mathcal{Y}[t_0 : T]$ given the observed history $\mathcal{Y}[0 : t_0]$ as measured by RMSE. The shaded areas represent 95% confidence intervals.

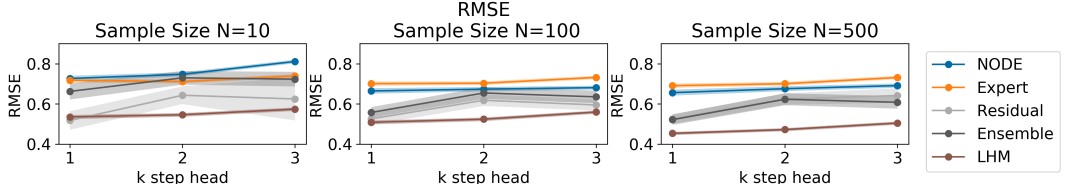

Figure 7: RMSE metric for each future time step. Predictions are issued based on historical measurements $\mathcal{Y}[0 : 12]$. The shaded areas represent 95% confidence intervals.

1. Learning rate: 0.01

2. Batch size: 10

3. Early stopping tolerance: 10 epochs

4. Max iteration: 400

5. Number of latent variables in NODE: $Z = E + M$, i.e. the true value. (additional settings $E + M + 4$ and $E + M + 9$ are reported in the sensitivity analysis)

6. Number of ML latent variables in LHM: $M$, i.e. the true value.

7. Latent dimensionality in Encoder: $2D$

8. Number of layers in NODE: 2

9. ODE Solver: dorip5

10. ODE rtol: 1E-7 (library default)

11. ODE atol: 1E-8 (library default)

### A.4.3 Performance under different lengths of observed history

As a reminder, we use the historical measurements $\mathcal{Y}[0 : t_0]$ up to some time $t_0$ to *predict* the future measurements $\mathcal{Y}[t_0 : T]$. To evaluate the performance under different lengths of observed history, we set $t_0 = 5, 10,$ or 12 days and use the default setting $\sigma = 0.2$ and $M = 2$. The results are presented in Figure 6, where each panel corresponds to a different $t_0$. As expected, the predictive performance improved when longer observed history is available. LHM outperforms the benchmarks for all $t_0$'s we study.

### A.4.4 Performance breakdown for each time step

In previous figures, we report the aggregated performance over the time horizon $t \in [t_0, T]$. To provide more details, here we report the metric for each time step. Specifically, we set $t_0 = 12$ and shows the performance of 1, 2, and 3 step ahead prediction. We consider the scenarios with different samples sizes for a typical setting (noise $\sigma = 0.2$ and $M = 2$ un-modeled latent variables $\mathbf{z}^m$). The RMSE metric is shown in Figure 7 and the CRPS metric is shown in Figure 8. As expected, the prediction error increases as the model predicts further into the future. LHM achieves best or equally best performance for each individual time steps

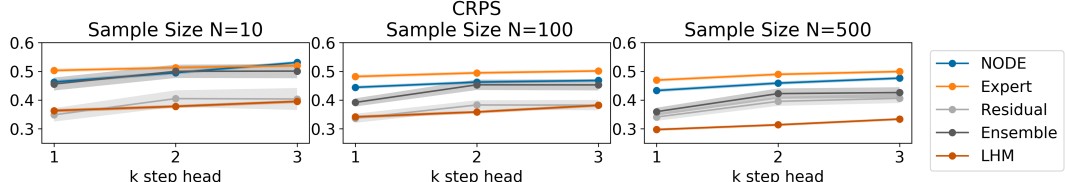

Figure 8: CRPS metric for each future time step. Predictions are issued based on historical measurements $\mathcal{Y}[0:12]$. The shaded areas represent 95% confidence intervals.

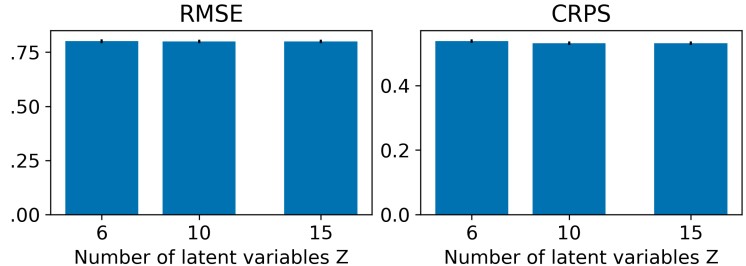

Figure 9: **NODE's performance under different numbers of latent variables** $Z$. The data are generated from six *true* latent variables. Prediction accuracy on future measurements $\mathcal{Y}[5:T]$ given the observed history $\mathcal{Y}[0:5]$ as measured by RMSE and CRPS.

### A.4.5 Performance of NODE is not sensitive to adding more latent variables

In the simulations reported above, we set the number of latent variables in NODE to be the true value, i.e. $Z = M + E$. In practice, $Z$ is a hyper-parameter that we do not know a priori. Here we study if the performance is sensitive to the exact choice of $Z$. We consider a setting where the data is generated from 6 latent variables (including both $\mathbf{z}^m$ and $\mathbf{z}^e$) and we vary $Z = 6, 10, 15$. We present the results in a typical simulation setting with $N_0 = 100$, $\sigma = 0.2$. As we show in Figure 9, the predictive performance does not significant change even when $Z$ is more than doubled. Note that similar findings have been reported in prior research [22]. This supports our choice of setting $Z = M + E$ by default.

### A.4.6 Performance gain with Normalizing Flows

To ensure a fair comparison with existing methods, we use the diagonal Gaussian distribution in LHM as the variational distribution. However, diagonal Gaussian is a restrictive approximation because it does not capture any correlation structure between the latent variables.

Here we study if using a more flexible distribution will lead to further performance gain. We adopt the planar normalizing flow proposed in [70] with the number of flows set to 4. As is standard in the literature, we amortize the initial conditions $\mathbf{z}(0)$ as well as the flow parameters $\mathbf{u}$, $\mathbf{w}$ and $\mathbf{b}$. The following shows a typical simulation with $N_0 = 100$, $\sigma = 0.4$ and $M = 2$.

Figure 10 tracks the loss function (negative ELBO) during training on the training and the evaluation data respectively. As we expected, the version with normalizing flow (LHM-NF) consistently achieves smaller loss on the training data due to the increased flexibility of the variational distribution. The improvement persists when we turn to the evaluation data, and eventually translates into the performance gain illustrated in Figure 7. This suggests that using a more flexible variational distribution (e.g. normalizing flow) tends to improve accuracy as well as the uncertainty estimation.

### A.4.7 Illustration of true and reconstructed trajectories

Figure 11 shows an observed trajectory ($\sigma = 0.2$, $M = 2$). A trained LHM takes the history up to $t = 12$ and reconstructs the whole trajectory for $t \in [0, 14]$. The reconstructed trajectory closely follows the trend of the true trajectory.

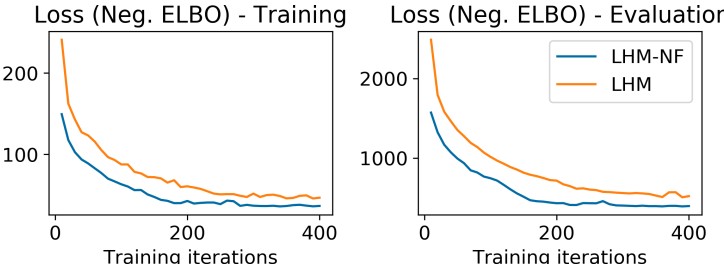

Figure 10: **Comparison between the standard LHM and the version with normalizing flow (LHM-NF)**. Loss on training and evaluation datasets are plotted over training iterations. The loss is the negative ELBO.

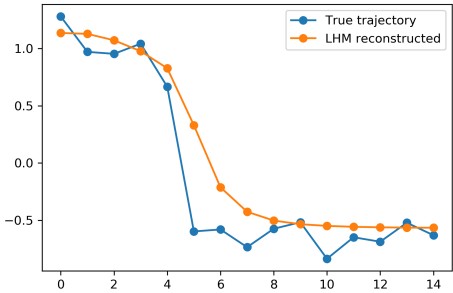

Figure 11: Illustration of true and reconstructed trajectories.

Table 2: Different methods to create hybrid ML models. We consider a static prediction problem with covariates $\mathbf{x} \in \mathbb{R}^D$ and target outcome $\mathbf{y} \in \mathbb{R}^K$ (notations differ from the rest of the paper). The $\mathbf{r} := \mathbf{y} - \hat{\mathbf{y}}$ denotes the residuals.

| Method | Example | Expert model | ML model | Final output |
|---|---|---|---|---|
| Residual Model | [54, 86] | $\hat{\mathbf{y}} = f^e(\mathbf{x})$ | $\hat{\mathbf{r}} = f^m(\mathbf{x})$ | $\hat{\mathbf{y}} + \hat{\mathbf{r}}$ |
| Ensemble | [93, 91] | $\hat{\mathbf{y}}_1 = f^e(\mathbf{x})$ | $\hat{\mathbf{y}}_2 = f^m(\mathbf{x})$ | $w_1\hat{\mathbf{y}}_1 + w_2\hat{\mathbf{y}}_2$ |
| Feature Extraction | [41] | $\mathbf{z}^e = f^e(\mathbf{x})$ | $\hat{\mathbf{y}} = f^m(\mathbf{z}^e)$ | $\hat{\mathbf{y}}$ |
| LHM | This work | Eq. 3 | Eq. 4, 5 | Eq. 4 |

### A.4.8 Computational resources

The simulations were performed on a server with a Intel(R) Core(TM) i5-8600K CPU @ 3.60GHz and a Nvidia(R) GeForce(TM) RTX 2080 Ti GPU. All individual simulations were finished within 3 hours.

## A.5 Extended related works

### A.5.1 Hybrid models

Table 2 categorizes various hybrid modeling frameworks in terms the kind of expert model and the kind of machine learning that are used. For illustrative purposes, we consider a static prediction problem with measurements (covariates) $\mathbf{x} \in \mathbb{R}^D$ and target outcome $\mathbf{y} \in \mathbb{R}^K$.

### A.5.2 Other research areas that involve ML and expert ODEs

There are several research areas that involve ML and expert ODEs but they are *unrelated* to hybrid model or LHM. We briefly describe them for clarification and completeness.

**Reduced-Order Models.** The expert model may involve a large number of variables, but not all of them are important to the system dynamics (e.g. in large, high-fidelity models of fluid dynamics [50]). Reduced-Order Models (ROMs) are compact representations of the more complex models [88]. They are often constructed using dimensionality reduction to retain only the most important dynamical characteristics of the original model. ROMs often achieve better estimation efficiency and lower the computational cost. Recently, ML has been applied to ROMs and achieved promising results [13, 90].

In ROMs, we start with an expert model that is *over-complete* and contains redundant variables. In contrast, in LHM, we are given a pharmacological model that is *incomplete*, i.e. it cannot fully explain the high-dimensional clinical measurements or provide the link between expert variables and the measurements. Hence, LHM is essentially solving the *opposite* problem of ROMs as we are introducing additional machine-learned latent variables into the system.

**Using ML to solve expert equations**. Some expert models involves ODEs or PDEs that are computationally challenging to solve (e.g. the quantum many-body problem [12]). ML has been used to speed up the solution process by making various approximations [36, 79]. However, the pharmacological models are generally well-behaved and the standard ODE solvers are able to find the solutions efficiently.

**Learning unknown ODEs from data** Step-wise regression is a general framework to discover unknown ODEs from data. It applies symbolic or sparse regression to the the observed time derivatives. When these time derivatives are not observed, they are first estimated from the (frequently-sampled) observations (e.g. by finite difference method) [10, 11, 74]. This approach is not applicable to our setting because the time derivatives of the expert variables $\dot{\mathbf{z}}^m$ are not observed or can be easily estimated from the data.

In addition to neural ODEs, Gaussian Processes (GP) have also been used to approximate unknown governing equations [5, 75]. However, most existing works focus on the discrete-time setting or use fixed step ODE solvers.

### A.5.3 Using Pharmacology/Biology models in ML

Several other works have proposed to integrate pharmacological models into machine learning. But the problem settings they considered and the approach they took is different from LHM.

[39] introduces a pharmacological model (the log-cell kill model) to modulate the state transition dynamics of a state-space model. Their work considers discrete-time dynamical systems rather than the continuous-time systems we focus on. The authors recognize that the existing log-cell kill models are inadequate to model the disease dynamics (e.g. failure to capture relapses). To address this shortcoming, the authors designed a new set of expert equations to allow for more complex dynamics before integrating them with ML. Hence, this approach requires a deep understanding of the expert model, and a fair amount of mathematical knowledge and manual work to modify the expert model. Furthermore, this modification process has to be repeated for a different expert model. In contrast, LHM learns the missing dynamics by introducing the ML latent variables $\mathbf{z}^m$ and neural ODEs $f^m$.

[94] considers a problem with more expert variables than observable physiological variables, which is opposite to the setting we consider.[4] The problem setting is similar to the reduced-order models discussed above. The authors use a neural network with time $t$ as input and outputs the system status at that time. In contrast, LHM uses neural ODEs to model the time derivatives and obtains the system status at time $t$ by solving the ODEs. Finally, the authors evaluate the gradient of the neural network with respect to $t$ by automatic differentiation and introduce an additional loss function to ensure the network gradient matches the expert ODEs. LHM does not involve any heuristic modification on the loss function and follows the standard practice in Bayesian inference.

### A.5.4 Causal treatment effect estimation

LHM predicts the disease progression based on governing ODEs. The causal inference literature studies related problems that involve predicting/estimating treatment effects. Although causal inference is a diverse field, most existing methods operate in the static or the discrete-time setting. In

---

[4]The Systems Biology models considered in [94] usually involve a large number of expert variables. This is not the case in the pharmacological models we consider.

Table 3: Methods that are related to predicting future outcomes given the intervention. LHM is based on the physical/mechanistic view on causality, which naturally applies to the continuous-time setting. Alternative approaches are developed based on different notions of causality but operate in static or discrete-time settings.

| Framework | Example | Temporality | Core component |
|-----------|---------|-------------|----------------|
| Physical/Mechanistic | LHM | Continuous-time | Governing equations (ODEs) of the dynamics |
| Potential outcome (PO) | [73] | Static / discrete | Statistical properties of the PO |
| Causal graphical | [65] | Static / discrete | Causal graphs (DAG) of the variables |
| Causal structural | [66] | Static / discrete | Structural equations between the variables |

contrast, LHM operates in the continuous-time setting due to the irregularity of clinical measurements. Here we compare the approach taken by LHM with other approaches in the literature for completeness. These methods are summarized in Table 3.

As discussed in Section 4, LHM predicts the future health status given treatments using the governing equations (ODEs). This corresponds to the mechanistic (or physical) notion of the causality, which is recognized as the "gold standard" for modeling natural phenomena by [76]. The governing ODEs describes the system dynamics in *continuous time*, which is essential because the clinical measurements are made at irregular time points.

The potential outcome framework widely used in Statistics is based on a different notion of causality [73]. It makes assumptions about the statistical properties of the unobservable potential outcomes (e.g. independence) to make inference about the (conditional) average treatment effect. Here, the focus is not on using or discovering the underlying governing equations, but on leveraging the statistical associations between the observed and the potential outcomes. Unlike the mechanistic framework, the potential outcome framework does not require the system to be observed over time, making it suitable for problems involving only static variables.

The causal graphical model [65] uses another notion of causality. It describes the causal structure between variables as a graph (typically a directed acyclic graph, DAG). Various identification strategies have been developed to infer the causal effect given the graph (e.g. the backdoor criterion [66]). A closely related framework is the structural causal model [66], where a set of structural equations are given in addition to the causal graph. Typically, the structural equations are standard equations that link the (static or discretely sampled) variables, but they are not ODEs that describe the continuous-time dynamics.

## A.6   Real data experiment

### A.6.1   List of clinical variables

We use the measurements of the following temporal physiological variables. These variables are chosen by our clinical collaborators and reflect the information accessible and important to a clinician when deciding the treatment plan. They include vital signals, lung mechanics, and the biomarkers measured in blood tests.

1. P/F ratio
2. PEEP
3. SOFA
4. Temperature
5. Arterial blood pressure
6. Heart Rate
7. Bilirubin
8. Thrombocytes
9. Leukocytes
10. Creatinine
11. C Reactive Protein
12. Arterial lactate

13. Creatine kinase

14. Glucose

15. Alanine transaminase

16. Aspartate transaminase

17. Prone positioning

18. Tidal volume

19. Driving pressure

20. FiO2

21. Lung compliance (static)

22. Respiratory rate

23. Pressure above PEEP

24. Arterial PaCO2

25. Arterial PH

26. PaCO2 (unspecified)

27. PH (unspecified)

We used the following static covariates:

1. Age

2. Sex

3. Body Mass Index

4. Comorbidity: cirrhosis

5. Comorbidity: chronic dialysis

6. Comorbidity: chronic renal insufficiency

7. Comorbidity: diabetes

8. Comorbidity: cardiovascular insufficiency

9. Comorbidity: copd

10. Comorbidity: respiratory insufficiency

11. Comorbidity: immunodeficiency

### A.6.2 Eligibility criterion

We selected all patients in DDW who stayed in the ICU for more than 2 days and less than 31 days (2097 out of 3464). Patients with a very short length of stay will not give us enough data points for training or evaluation.

### A.6.3 Hyper-parameter settings

The following is the hyperparameter setting used in the real-data study. They are decided based on a pilot study.

1. Learning rate: 0.01

2. Batch size: 100

3. Early stopping tolerance: 10 epochs

4. Max iteration: 1500

5. Number of latent variables in NODE: 20

6. Number of ML latent variables in LHM: 15 (this is to ensure the total number of latent variables is the same as NODE).

7. Latent dimensionality in Encoder: $1.2D$

Table 4: Prediction accuracy (RMSE) on $\mathcal{Y}[t_0 : t_0 + H]$ over different time horizons $H$ (hours). The standard deviations are shown in the brackets.

| Method \H= | 6 | 12 | 24 | 72 |
|---|---|---|---|---|
| Expert | 0.734 (0.99) | 0.724 (1.00) | 0.713 (0.03) | 0.993 (0.03) |
| Residual | 0.555 (0.98) | 0.575 (1.08) | 0.607 (0.04) | 0.983 (0.05) |
| Ensemble | 0.556 (0.71) | 0.573 (0.73) | 0.599 (0.04) | **0.713 (0.05)** |
| NODE | 0.661 (1.00) | 0.654 (1.00) | 0.650 (0.02) | 0.996 (0.02) |
| ODE2VAE | 0.627 (1.11) | 0.616 (1.09) | 0.619 (0.02) | 1.113 (0.01) |
| GRU-ODE | 0.549 (0.71) | 0.571 (0.72) | 0.601 (0.04) | **0.711 (0.05)** |
| Time LSTM | 0.610 (0.81) | 0.620 (0.82) | 0.631 (0.04) | 0.807 (0.05) |
| LHM | **0.517 (0.72)** | **0.511 (0.73)** | **0.511 (0.03)** | **0.691 (0.03)** |

8. Number of layers in NODE: 2

9. ODE Solver: adams

10. ODE rtol: 1E-7 (library default)

11. ODE atol: 1E-8 (library default)

### A.6.4 Accuracy over different time horizons

Table 4 shows the performance over different prediction horizons $H$ given $N_0 = 1000$ training samples. LHM achieves the best or equally the best performance in all cases.

### A.6.5 License and anonymity

Access to the DDW is regulated. We have signed an end user license before access to the data was granted. All data were pseudonymized in DDW.

### A.7 Practical extensions

**Incorporating static covariates.** Static covariates such as the demographics often impact disease progression. We can easily incorporate these variables in LHM by treating them as time-constant "treatments". This will allow the static covariates to impact the latent dynamics as well as the mapping between the latent and physiological variables (Equation 3 to 5).

**Informative sampling.** It is well known that the sampling frequency may carry information about the variables being measured (e.g. clinicians tend to take measurements more often if a patient is critically ill) [3]. One approach to incorporate informative sampling is to explicitly model it as a marked point process [72]. Another popular approach is to concatenate the measurements $\mathbf{x}$ with the masking vector that indicates which variable is measured, and train the model on the extended measurement vector [45]. Both approaches are compatible with LHM.

**Correcting model mis-specification.** *Equation Replacement* is a general approach that applies to any *misspecified* expert model and it can be combined with all the methods discussed above, and to LHM [35, 64, 98]. In this approach, one first identifies which equations in the expert model are misspecified, and then replaces these by flexible function approximators (such as neural networks), that will approximate the true equation after training. Equation replacement only attempts to correct the misspecifications in the original model, but does not introduce any new variables.

**Efficient online inference.** The inference method presented in Section 3.4 requires to re-process the entire history each time a new measurement is made. Instead, it may be desirable to incrementally update the posterior of $\mathbf{z}_i(0)$ based on the most recent measurement only. Fortunately, online Bayesian update (also known as Bayesian Filtering) is a well studied problem with many proven solutions (e.g. Kalman filter and extensions [71]). These inference methods can be applied when the efficiency of online inference is of concern.

**Improving encoder architecture.** For a fair comparison with related works, we used the reversed time-aware LSTM encoder proposed in [14]. Essentially, it is a LSTM with the observation time as an additional input channel and running backward through time. To further improve performance, one may explore other architectures. Essentially, any architecture that takes irregularly sampled data

as input is applicable. Examples include the Neural Controlled Ordinary Differential Equation [45] and the Neural ODE Processes [61].