# OpenReview forum: "Integrating Expert ODEs into Neural ODEs: Pharmacology and Disease Progression"
_NeurIPS.cc/2021/Conference — NeurIPS 2021 Poster_

### Official Review · Reviewer_DE5k · 2021-07-06

**Rating:** 8
**Confidence:** 3

**Summary:**

This paper proposes a novel hybrid modeling framework, the Latent HybridisationModel (LHM), that imbeds a given pharmacological model (a collection of expert variables and the ODEs that describe the evolution of these variables ) into a larger latent variable ML model (a system of Neural ODEs).
Nice approach and Extendable to other fields and applications

**Ethical Concerns:**

No ethical Issues

**Limitations And Societal Impact:**

Some Limitations are presented in the manuscript  but they have not a special chapter or paragraph. It is always the same issue of unorthodox structure that limits the reader and the  full appreciation of the ideas. I kindly recommend to revisit your manuscript's structure and create a Limitation paragraph (like the medical articles) inside the Discussion chapter, for both methodological and population/simulation/experimental limitations.

**Main Review:**

Originality:
Methods are not new, but there is a novel combination of previous techniques. It is clear how this approach is distinguishable from the others and most of the related works are sufficiently cited

Quality: I
As a strength,  the theoretical grounding and the empirical evaluations are sufficient and they provide nice details about both simulations and real datasets. I would say that I found the approach very interesting and towards the appropriate control/exploitation of the expertise (pharmacological or other) by the models.
It seems a rather complete work.

As a weakness, I would mention the manuscript structure. I do not understand the reason that authors, although they have some clear ideas, they chose a rather “unorthodox” manuscript structure in my opinion.
Briefly,
1.	I would propose Related works before Methodology
2.	3.5 subchapter seems more like a Discussion part
3.	I would propose enrichment in the Discussion part… Why LHM works better than the others, where the others make it better and in which conditions. Mostly take some already written arguments and elements from the manuscript and create a structured discussion chapter.

Clarity: Is the submission clearly written? Is it well organized? (If not, please make constructive suggestions for improving its clarity.) Does it adequately inform the reader? (Note that a superbly written paper provides enough information for an expert reader to reproduce its results.)
ideas are clearly presented, I have already made the comments about the structure in order to avoid repetitions.

Significance:
The results are important and such method may help not only practitioners but also other fields in creating personalized models(from the clinicians point of view) that are related to their daily practice.

**Time Spent Reviewing:**

2

---

> ### Author Response · Authors · 2021-08-10
> **Response to Reviewer DE5k**
>
> Thank you for your thoughtful comments and suggestions.
>
> In the revision, we will reorganize the sections and expand the discussion to further improve readability.

---

### Official Review · Reviewer_V3vo · 2021-07-11

**Rating:** 5
**Confidence:** 4

**Summary:**

This paper presents a new model called latent hybridisation model (LHM) to integrate a system of expert-designed Ordinary Differential Equations (ODEs) -- from domain knowledge, with Neural ODEs -- learnt from data. The aim is to enable learning in small sample regimes and have interpretable latent variables. Empirical results are shown on a clinical task of predicting temporal physiological variables after treatment, using a pharmacological (expert) model of the treatment effect.

**Ethical Concerns:**

None, to my knowledge.

**Limitations And Societal Impact:**

Societal impacts have been adequately addressed in section A.2.

Limitations: briefly mentioned in section 6 but could be elaborated further. What is mentioned now is the current modeling inadequacy of the model. It's limitations with respect to its performance and other existing models needs to be described in more detail. This can also be done empirically (as discussed earlier).

**Main Review:**

My comments along the 4 dimensions:

1. Originality

The task/problem addressed is well known. It is clinically important and challenging.

The model appears, to me, as a new combination of older known techniques. It is well developed for this application. Previous related work has been adequately cited but the explanation of how the proposed model differs from previous models could be improved further.

2. Quality

The model and its inference appears to be sound. However, the experimental results could be further improved to better support the claims and provide more insights into the strengths and weaknesses of the model.

- For example, in both the simulation and real settings, only one-time treatment has been considered. In dexamethasone treatment the dose can be once daily and multiple time points could be considered along with the experimental evaluation. In other applications this may be even more important to consider.

- Simulation settings could be further expanded, e.g., by varying values of T and characteristics of individual time series (to model clinical data).

- I believe the results show the RMSE aggregated for all the physiological variables (both simulations and real data). Are some variable's predictions worse than others? In real clinical data, the variables differ in data type and frequency of measurement. How do the predictions differ with respect to these characteristics? Model performance could be analyzed along these dimensions.

- Similarly the impact of expert knowledge could also be quantified better. As a practitioner, one would want to know when to use a particular expert model, and if there are multiple models, which ones to choose?

- Is the improved model performance due to the expert model added or is it just because LHM has more latent variables/capacity compared to other models compared? An experiment could be suitably designed to answer this, e.g., by comparison with another hybrid model where an alternative "false" model of similar capacity has been added. I think this is important to convince the reader of the utility of the expert model.

- Real data results are shown on only 1 dataset. More datasets (that are available, for other treatments/diseases) could be added.

- Is there some way to evaluate the correctness of the results in fig. 4, rightmost column? Perhaps through some proxy clinical variable. Similarly, to convince the reader of the interpretability/accuracy of all the expert latent variables, suitable comparisons could be presented.

- What are the disadvantages of LHM, compared to existing models? Does it require more training time? Does it get biased by the expert model - if so, how? Under what circumstances would that lead to worse results?

3. Clarity

The paper is clear, very well written.

4. Significance

The model is potentially useful in practice but its advantages and limitations need more assessment (as described above). It addresses two challenging problems (small samples, interpretability) in the clinical context and develops a good solution for them, as shown in the preliminary experimental results. In my opinion, the work is promising but incomplete.



**Time Spent Reviewing:**

5 hours

---

> ### Author Response · Authors · 2021-08-10
> **Response to Reviewer V3vo**
>
>
> Thank you for your thoughtful comments and suggestions. We’ve provided a point-by-point response below. Please let us know if anything needs further clarification.
>
> ---
>
> ### (1) Treatment settings and time horizons in the experiments
>
> Actually, we have considered multiple treatments. In our experiment using real data, we used the actual treatments received by the patient. Approximately 30% of the patients in the dataset received more than one treatment. In the revision, we will emphasize this point and include the summary statistics of the treatments.
>
> In the real data experiments, we have also considered multiple prediction horizons. The results are presented in Table 4 in the Appendix. We will add a pointer to this result in the main text.
>
> ### (2) Breakdown of the performance metric
>
> We reported the aggregated metric because we’ve seen a consistent improvement across different physiological variables. In the revision, we will provide a breakdown of the performance metric for each physiological variable.
>
> ### (3) Choice of expert model
>
> In the current work, we attempt to incorporate one expert ODE. When multiple candidate ODEs are available, we can perform model selection (e.g. cross validation) to decide which ODE leads to better prediction accuracy. Incorporating multiple expert ODEs in the model is an interesting area for future research (Section 6).
>
> ### (4) Performance gain and model capacity
>
> To clarify: the NODE baseline has *at least* as many latent variables as LHM in the experiments (line 265). The exact number is decided by hyper-parameter tuning on the validation data. This is to ensure that NODE has equal (or higher) capacity than LHM.
>
> NODE has *more* parameters than LHM. This is because LHM uses an expert ODE with few parameters while NODE relies solely on neural networks.
> Hence, the performance gain of LHM is not due to adding capacity or parameters; in fact, LHM has lower capacity and fewer parameters.
>
> ### (5) Evaluation of the case study
>
> In this study, we have relied on ICU physicians with experience in treating COVID-19; they have qualitatively verified the variables in Figure 4. We did not acknowledge any contributing clinicians in the current version because of the anonymity requirement, but we will do so in the final version.
>
> ### (6) Limitation of LHM
>
> LHM starts with an expert ODE (assumed to be correct) that models the dynamics of the expert variables. Thus, it cannot be applied to cases where the expert ODE is not available. The user of LHM also needs to ensure that the physiological variables $x(t)$ are related to the expert variables $z^e(t)$ (i.e., that Equation 4 holds). Otherwise, it will be impossible to infer the expert variables from the measurements.

---

> > ### Comment · Reviewer_V3vo · 2021-08-31
> > **Acknowledgement**
> >
> > Thank you for your responses.

---

> ### Author Response · Authors · 2021-08-25
> **Dear Reviewer V3vo**
>
> Once again, thank you for your invaluable feedback. We were wondering whether our response has addressed your concerns. If you have any additional comments, please let us know, we would be eager to address them.

---

### Official Review · Reviewer_TPwF · 2021-07-12

**Rating:** 4
**Confidence:** 5

**Summary:**

This paper proposes a latent hybridization model that incorporates pharmacological knowledge into ML models. In particular, it utilizes expert pharmacological variables and regular clinical measurements to improve the optimization process of neural ODEs. The model was evaluated on a synthetic dataset and a real-world COVID-19 dataset.

**Limitations And Societal Impact:**

No.

**Main Review:**

Pros:
1.	This idea of leveraging pharmacological models in ML modeling makes sense. The clinical analysis for the model is also interesting
2.	The performance on COVID-19 dataset looks good, especially under small dataset size.

Con:
1.	There are some unclear assumptions. For example, Eq. 6 assumes the independency between two measurements Y(t0:T) and Y(0:t0) for patient i. Does this discard the temporal dependencies in clinical measurements?  This is important since the paper claims to model a system's temporal behavior.

2.	The paper makes a broader claim but only demonstrates a very narrow application. For example, it claims that the model integrates pharmacology model into neural ODEs for "predicting the progression of disease under medications". Usually when we claim a model is designed for predicting disease progression, we need to at least evaluate its utility against chronic diseases or neurodegenerative diseases that actually have longer term progression. However in this paper, the model is only evaluated on two COVID-19 dataset (synthetic and real-world).  It is unclear whether the model would outperform existing models on real disease progression tasks against state-of-the-art baselines. Therefore, without major improvement to demonstrate the general utility, the paper is just a narrow-focused application paper, rather than what it claims.

3. The experiments are weak due to missing many important baselines.  The authors only compare with Time LSTM as the non-ODE model, which is not persuasive considering there are lots of more recent DL-based disease progression models, as well as DL-based disease progression/prediction models that incorporate expert rules/knowledge.




**Time Spent Reviewing:**

2

---

> ### Author Response · Authors · 2021-08-10
> **Response to Reviewer TPwF**
>
>
> Thank you for your thoughtful comments and suggestions. We’ve provided a point-by-point response below. Please let us know if anything needs further clarification.
>
> ---
>
>
> ### (1) Positioning
>
> We regret that the original manuscript did not seem to make clear the intended objective of our work and position our work with respect to other work.
>
> Our work does *not* intend to achieve state-of-the-art performance in modeling the progression of any specific disease. Rather, our work intends to bridge the gap between the laboratory and the clinic.
>
> We start with a set of expert variables $z^e$ which are measurable in the *laboratory* and an expert ODE that *correctly* models the dynamics of these variables.  (This expert ODE may have been the result of decades of scientific research and numerous laboratory experiments.).
>
> If the expert variables were observed/measured in the *clinical* environment, there would be nothing for us to do.  However, in many situations, these expert variables $z^e$ are *not* observed/measured in the clinical environment.  This may be because the measurement process would be costly, or time-consuming, or invasive.  Hence, to leverage the scientific insight encoded in the expert ODE, we need link between the expert variables to measurements that *are* made in the clinical environment.  LHM does so by introducing various ML components.
>
> We will make this much clearer in the revision.
>
>
> ### (2) Temporal dependency
>
> In Equation 6, the independence between $Y[0:t_0]$ and $Y[t_0, T]$ is *conditioned* on the *latent* variables $z$. Without the conditioning, there would be (marginal) temporal dependency between $Y[0:t_0]$ and $Y[t_0, T]$.

---

> ### Author Response · Authors · 2021-08-25
> **Dear Reviewer TPwF**
>
> Once again, thank you for your invaluable feedback. We were wondering whether our response has addressed your concerns. If you have any additional comments, please let us know, we would be eager to address them.

---

### Official Review · Reviewer_fnTo · 2021-07-14

**Rating:** 4
**Confidence:** 5

**Summary:**

The contribution of the paper is twofold. On one hand a hybrid model to incorporate background knowledge in a form of ODEs to a Neural ODE model is presented. On the other hand a causal inference tool was provided for time series data. The work focuses on interpretability and clinical decision support.

**Limitations And Societal Impact:**

The work clearly have societal impact as it  addresses clinical decision support, this is described in the paper. As the work also describes, the expert ODE integration is limited for self contained expert odes .
However, the causal assumptions are not well described. The following questions need to be addressed: Does the method assume "no unobserved confounders"? Does it assumes (strong)ignoraebility?  Does it assume missing at random observation? Are these assumptions at least approximately valid in the application domain?


**Main Review:**

On integrating expert knowledge:
The hybrid model is theoretically solid and the assumptions are well presented. However, due to the problem of confounding in real world setting (discussed below) the assumption of autonomous evolution of the expert model is likely violated even if there is no direct feedback from $z^m$ to $z^e$, as there will be a feedback loop through $a$, as the clinician likely use information from the measurements to select the next treatment.

There have been a work on ODE based expert knowledge integration in this year ICLR: Le Guen, Vincent, et al. "Augmenting physical models with deep networks for complex dynamics forecasting." Ninth International Conference on Learning Representations ICLR 2021. 2021. (ICLR2021 is in the 2 month time window before submission deadline therefore this work is contemporaneous)

On causal inference:
There are major issues with the handling of causality.
The data is not sporadically observed but rather a sequence of labelled evens. It is not true that the measurements are missing at random. The fact that the physician order a measurement already indicate something on their belief (the missingness indicator causally depend on some hidden state).

The fact that the model estimate individual treatment effects (conditioning on individual initial state $z_i(0)$) places the model to rank 3 on Pearl's ladder of causation, and require operations on counterfactuals.

The model assumes no unobserved confounders, what is a strong assumption in this particular case, when COVID-19 treatments were followed up in a non-controlled experiment.
All actions are considered exogenous (there is no backdoor path) as shown on Figure 1. C). This is even stronger assumption than "no unobserved confounders" and definitely violated by any dataset recorded in clinical practice. This assumption would mean the physician do not use any information in the previously measured variables when deciding on the next treatments what we passively observe.
Even more complicated dependence structures can exist in real life sequential treatment situations, like the kind addressed by [ J. Pearl and J.M. Robins. Probabilistic evaluation of sequential plans from
causal models with hidden variables.  Uncertainty in Artificial Intelligence 11, 1995, p 444–453. ]
Please define all the assumptions necessary for your method to estimate unbiased treatment effects, and provide a sufficient argument that the dataset in the Experimental section indeed satisfy these conditions.

Real-data experiments: The illustration of clinical decision support is not possible to verify. The paper contains counterfactual statements based on the prediction but there is no evidence these statements are indeed valid. Only reference we have is a possibly heavily confounded observational data. To make claims on value for estimating individual treatment effects like "LHM predicts that immune activity $z^e_1$ would decrease afterwards even without treatment. " we need some experimental data, or a detailed simulation.



**Time Spent Reviewing:**

5

---

> ### Author Response · Authors · 2021-08-10
> **Response to Reviewer fnTo**
>
> Thank you for your thoughtful comments and suggestions. We’ve provided a point-by-point response below. Please let us know if anything needs further clarification.
>
> ---
>
> ### (1) Causal inference
>
> We believe that the reviewer has misunderstood a key point of our work. LHM is not developed as a “causal inference” algorithm, and it does not estimate ‘treatment effects or “counterfactuals” either in the Neyman-Rubin framework or the Pearl framework. The differences are discussed in Appendix 5; we will add a pointer to this discussion in the main text. (Note that term “causal inference” does not even appear in the main text.).
>
> LHM makes use of a (correct) expert ODE --- given the current state value and the future actions, the ODE is solved to give future state values. By *changing* the actions, the ODE will give different future state values.
>
> We reiterate that the expert variables correspond to quantities that are measurable in the laboratory (Figure 1). Scientists may conduct controlled experiments in the lab to discover and validate the expert ODEs. However, this work does not concern the problem of discovering expert ODEs from experimental data --- we use an expert ODE that has been validated.
>
> The machine learning components and ODEs are introduced because the expert variables are not observable in the clinical setting, and so must be inferred from what can be observed in the clinical setting.  We do not claim that the machine-learned latent variables necessarily correspond to physiologically meaningful quantities.  If they do, that might be useful – but it is not necessary for our purpose, which is to infer the unobservable expert variables.
>
> ### (2) Related work
>
> Thanks for bringing our attention to the contemporaneous work [1]. We will cite and discuss this work in the revision, but we believe our work and [1] address very different problems and use very different methods.  Our work is not subsumed by [1], nor is it a small variation on [1].
>
> The work [1] assumes that the expert ODE $f^e$ is *incorrect*. To address this issue, [1] introduces a ML component $f^m$ to *additively* correct for the discrepancy between the expert ODE and the true dynamics $f$, i.e. $f^m = f - f^e$. It is thus related to the residual models discussed in Section 4.
>
> In contrast, our work assumes the expert ODEs are *correct*. This is a reasonable assumption in our application because the expert variables are measurable in the *laboratories*. Prior lab experiments and research may have established the correct ODE that governs the expert variables. However, the expert variables are not observable in the clinical setting.  Our key objective is to uncover the expert variables from the clinical measurements, thereby bridging the gap between the lab and the clinic. Our work does not attempt to modify or adjust an incorrect expert ODE (line 117-123). Rather, we construct additional ODEs that allow us to infer the unobservable expert variables $z^e$.
>
>
>
>
> ### (3) Experiments
>
> In the experiments, we do not evaluate the accuracy of predictions on “counterfactuals”. In the simulation study, the treatments and outcomes are *assigned* in both training and testing sets (line 230). In the real data study, we use the treatments that the patients *actually received* and evaluate the *factual* outcomes. We regret if this was not clear in the original manuscript; we will make it clearer in the revision.

---

> ### Author Response · Authors · 2021-08-25
> **Dear Reviewer fnTo**
>
> Once again, thank you for your invaluable feedback. We were wondering whether our response has addressed your concerns. If you have any additional comments, please let us know, we would be eager to address them.

---

### Decision · Program_Chairs · 2021-09-28

**Decision:**

Accept (Poster)

**Comment:**

This work integrates a system of expert-designed ODEs with a machine-learned neural ODEs to improve the prediction of disease progression. While the paper generated much discussion, ultimately shortcomings related to issues around causal inference from observational data remain. While the authors claim that they are not promoting this work as a causal inference algorithm, nonetheless the text states in Section 3.5 that the approach "can provide clinicians with prediction of the disease progression given the treatments, enabling the clinicians to design the best treatment plan for the patient at hand," which sounds a lot like estimating counterfactuals. Thus significant reframing seems necessary. I encourage the authors to carefully consider reviewer feedback, in particular with respect to the framing and empirical results, when working on their revision.

**Consistency Experiment:**

NeurIPS has a long history of experimentation. In 2014, NeurIPS ran an experiment in which 10% of submissions were reviewed by two independent committees to quantify the randomness in the review process. This year, we repeated a variant of this experiment to see how the quality of the review process has changed over time.  This paper was part of the experiment and was therefore assigned to two committees (consisting of reviewers, an Area Chair, and a Senior Area Chair) that reached independent decisions.  If both committees made the same recommendation, this recommendation was followed. If a single committee recommended acceptance, the paper was accepted (with the exception of a few cases in which the other committee identified what we considered a fatal flaw, e.g., an error in a key result).

This copy’s committee reached the following decision: **Reject**

The other committee assigned to the paper recommended **Accept (Poster)**.  You can find the other set of reviews, along with any follow up discussion with the authors here:
https://openreview.net/forum?id=tDqef76wFaO